# Decoding Game: On Minimax Optimality of Heuristic Text Generation Strategies

**Sijin Chen,**[*] **Omar Hagrass,**[†] **Jason M. Klusowski**[†]
Princeton University, Princeton, NJ 08540, USA
{chensj,oh2588,jason.klusowski}@princeton.edu

## Abstract

Decoding strategies play a pivotal role in text generation for modern language models, yet a puzzling gap divides theory and practice. Surprisingly, strategies that should intuitively be optimal, such as Maximum a Posteriori (MAP), often perform poorly in practice. Meanwhile, popular heuristic approaches like Top-$k$ and Nucleus sampling, which employ truncation and normalization of the conditional next-token probabilities, have achieved great empirical success but lack theoretical justifications. In this paper, we propose Decoding Game, a comprehensive theoretical framework which reimagines text generation as a two-player zero-sum game between Strategist, who seeks to produce text credible in the true distribution, and Nature, who distorts the true distribution adversarially. After discussing the decomposibility of multi-step generation, we derive the optimal strategy in closed form for one-step Decoding Game. It is shown that the adversarial Nature imposes an implicit regularization on likelihood maximization, and truncation-normalization methods are first-order approximations to the optimal strategy under this regularization. Additionally, by generalizing the objective and parameters of Decoding Game, near-optimal strategies encompass diverse methods such as greedy search, temperature scaling, and hybrids thereof. Numerical experiments are conducted to complement our theoretical analysis.

## 1 Introduction

Decoding strategies underpin the mechanism of generating a text sequence from a given language model, and therefore become an essential component of modern Large Language Models (OpenAI, 2024). Specifically, given an autoregressive language model $\widehat{\mathbb{P}}$ which encodes the conditional next-token probability $\widehat{\mathbb{P}}(X_t|X_{<t})$, one aims to generate a high-quality sequence $(X_1, \ldots, X_T)$ by some strategy based on $\widehat{\mathbb{P}}$. Perhaps one of the most straightforward strategies is Maximum a Posteriori (MAP), looking for the most probable sequence, i.e., the one with the maximum predicted likelihood $\widehat{\mathbb{P}}(X_1, \ldots, X_T)$. Considering the computation cost of an exact MAP, one would naturally turn to some local heuristic variants such as greedy search, beam search (Graves, 2012; Sutskever et al., 2014), and contrastive search (Su et al., 2022).

These deterministic searching methods based on likelihood maximization can achieve state-of-the-art performance especially in closed-ended tasks, such as translation, coding, math problem solving, and summarization (Shi et al., 2024). Counter-intuitively, in open-ended text generation tasks, these strategies usually lead to low-quality, degenerate texts, even with heavily trained state-of-the-art language models (Shi et al., 2024; Wiher et al., 2022; Hashimoto et al., 2019). Instead, stochastic sampling methods that randomly select the next token are observed to yield better outputs. Popular strategies include Top-$k$ sampling (Fan et al., 2018), Nucleus (Top-$p$) sampling (Holtzman et al., 2020), $\eta$ sampling (Hewitt et al., 2022), and Mirostat sampling (Basu et al., 2021), among others.

They follow a truncation-normalization design, namely (1) sampling the next token from a truncated distribution by removing the tail probabilities, and (2) rescaling the remaining probabilities by a normalizing constant. Some other methods like Basis-Aware sampling (Finlayson et al., 2024) and

---

[*]Department of Electrical and Computer Engineering
[†]Department of Operations Research and Financial Engineering

Typical sampling (Meister et al., 2023) also rely on truncation but may discard high-probability tokens besides the tail; see Section 2 for details. Formally, if $(\hat{p}_1, \ldots, \hat{p}_d)$ is the vector of the predicted probability of all candidate next-tokens $\{1, \ldots, d\}$, these methods decide an index set $\mathcal{S}$ and sample a token $i$ with probability

$$q_i \propto \hat{p}_i \mathbb{1}_{(i \in \mathcal{S})}.$$

Here, different designs define the truncation set $\mathcal{S}$ using distinct criteria: Top-$k$ sampling uses a fixed size threshold, Nucleus sampling uses cumulative probability mass, and entropy-based methods use information-theoretic thresholds ($\eta$).

Regarding decoding strategies, there is an interesting dichotomy between theory and practice. From a statistical perspective, likelihood maximization approaches are desired to succeed by seeking or approximating the posterior mode of $\widehat{\mathbb{P}}$, but usually underperform in practice on open-ended generation tasks. On the contrary, despite their empirical superiority over likelihood maximization, the design of randomized sampling strategies remains mostly heuristic and the theory behind is poorly understood. To resolve this dichotomy, this paper aims to propose a comprehensive theoretical framework of text generation, where *heuristic sampling strategies, rather than likelihood maximization, are proved to be (near-)optimal*.

Now, we shall introduce the motivations behind our framework before presenting it formally.

## 1.1 MOTIVATION AND OUR FRAMEWORK

**First thought.** At first sight, a statistician may naturally relate these truncation methods with the concept of sparsity and regularization, and further attempt to handcraft a constrained or penalized optimization problem where they are optimal strategies. This is easier than it may sound: for example, we can design a distance metric so that Top-$k$ sampling is the best sparse approximation to the original distribution according to this metric, such as $\ell_0$ distance that directly controls sparsity. The major flaw of such approaches is that their objectives and constraints mostly come from non-principled, reverse engineering and lack statistical motivations. Therefore, they may not be able to provide theoretical insight into questions like why sparse solutions are favored, and why we should adopt a specific regularization term and distance metric.

**Second thought.** Let us restart from the most significant observation that likelihood-maximization approaches fail in practice. What does this imply? If $\mathbb{P}$ is the *true* distribution of natural language, it is reasonable to expect that the trained language model $\widehat{\mathbb{P}}$ is away from $\mathbb{P}$. This makes $\widehat{\mathbb{P}}$-likelihood an unreliable criterion of a generated text, and hence leads to the failure of likelihood maximization. On the other hand, the appropriate criterion a strategy would like to maximize is the $\mathbb{P}$-likelihood of a generated sequence.

However, note that for generality, we restrict ourselves from assuming too many structures on the true distribution $\mathbb{P}$, except for its bounded deviation from $\widehat{\mathbb{P}}$. This "model-free" setup brings an adversarial nature to text generation: in the worst case, $\mathbb{P}$ can try its best to degrade the quality of our generated text within its distance budget.

**Decoding Game.** These ideas lead to our proposal, *Decoding Game*, a two-player zero-sum game between Strategist (S) and Nature (N). In this game, player S chooses a (randomized) decoding strategy to generate a text sequence that, in expectation, achieves good log-likelihood in the true distribution. On the other hand, player N is always able to shift the true distribution adversarially to reduce the text quality. Knowing the decoding strategy beforehand, player N chooses the worst-case true distribution $\mathbb{P}$. Formally, a $T$-step Decoding Game is represented by

$$\max_{\mathbb{Q}} \min_{\mathbb{P} \in N(\widehat{\mathbb{P}})} \mathbb{E}_{\mathbb{Q}} \log \mathbb{P}(X_1, \ldots, X_T \mid X_0),$$

where, conditioned on a given prompt $X_0$, $\mathbb{Q}$ is the probability measure on $(X_1, \ldots, X_T)$ induced by the decoding strategy of player S, and $N(\widehat{\mathbb{P}})$ refers to a neighborhood of $\widehat{\mathbb{P}}$. The objective of the game is the true log-likelihood of a length-$T$ sequence generated from strategy $\mathbb{Q}$, in expectation.

For player S, an equivalent perspective is robust optimization (Ben-Tal et al., 2009). Since player S has no knowledge of $\mathbb{P}$, it aims to find a strategy that can work consistently well for all the possible true distributions. To achieve such robustness against an adversarial distributional shift, player S

should optimize the log-likelihood in the *worst* case of $\mathbb{P}$ to ensure that its strategy does not lead to poor performance in *any* case, which corresponds to the minimax formulation.

If there is no adversary ($\mathbb{P} = \widehat{\mathbb{P}}$ always holds), then the game reduces to $\widehat{\mathbb{P}}$-likelihood maximization, and naive MAP is exactly the solution. However, when an adversary is present, MAP becomes sub-optimal and the game invites more interesting consequences.

## 1.2 CONTRIBUTION

We conduct in-depth investigations into Decoding Game. The main results include:

- **Criticality of one-step Decoding Game.** In Section 3, we identify the role of one-step Decoding Game in understanding the multi-step setting. We build up a recursive structure for the multi-step game, and argue the computational intractability of obtaining a global solution in modern LLMs. Instead, we construct a locally optimal mechanism that involves solving a one-step Decoding Game locally at each timestep, and justify its worst-case performance.

- **Optimality of heuristic strategies under implicit regularization.** Under total variation (TV) distance, we provide closed-form solutions to the one-step Decoding Game for both players. In Section 4.1, we show that the optimal strategy of player N imposes an $\ell_\infty$-type regularization on the log-likelihood. As a result, tail truncation-normalization sampling strategies emerge as first-order approximations to the optimal strategy of player S; see Section 4.2.

- **Generalizability of the framework.** In Section 4.3, we further discuss the consequences of using different objectives for Decoding Game, recovering other types of strategies such as temperature-based methods. The exclusive advantage of log-likelihood, in contrast to other objectives, is also highlighted under the general framework.

- **Empirical evidence.** Building on the general theory, in Section 5, we propose *Game sampling* (Algorithm 1) and empirically evaluate its performance in open-ended text generation with GPT-2 models. The experiments suggest that Game sampling is able to outperform other strategies, which corroborates our optimality results. The code is available at `https://github.com/omar-hagrass/decoding-game`.

Decoding Game provides a comprehensive theoretical framework that rigorously establishes optimality results for heuristic sampling strategies. It largely differs from existing interpretations which, to different extents, provide some theoretical viewpoints as partial justifications for their design. We will briefly review them in Section 2.

At the same time, we believe that the statistically meaningful motivations and minimal assumptions behind Decoding Game open up its potential for future research on decoding strategies; see discussions in Section 6.

## 2 RELATED WORKS

### 2.1 EXISTING THEORETICAL INTERPRETATIONS

Theoretical explanations for decoding methods have been very sparse and existing works are relatively limited. Known perspectives presented in literature include (1) overestimation of token probabilities, (2) surprisal and perplexity of generated text, and (3) implicit regularization on MAP.

Following the long-held intuition (Holtzman et al., 2020) that language models tend to assign excessive probability to the unreliable tail, Finlayson et al. (2024) explained truncation as a remedy for this problem, showing that it can correctly discard the tokens out of the support of the true distribution when overestimation is upper-bounded. They further credited overestimation to the Softmax Bottleneck (Yang et al., 2018) brought by the language model architecture, motivating a new truncation mechanism that may remove high-probability tokens besides the tail. However, this structural assumption may not well account for the broadness of the source of overestimation. Additionally, it is unclear why rescaling all the remaining probabilities by the same constant is considered the best approach after truncation.

A similar idea of identifying the correct support has also driven prior works such as Hewitt et al. (2022) and Meister et al. (2023). Hewitt et al. (2022) modeled the predictions as a mixture of true distribution and uniform-like smoothing distribution, and viewed truncation as a way to desmooth the output. Meister et al. (2023) proposed to compute the support that better aligns with the information-theoretic metrics of human text measured by token surprisal and entropy, under assumptions on the behavior of human speakers. In this method, high-probabilty tokens may be discarded as well.

Basu et al. (2021) theoretically derived the perplexity of various sampling methods under the statistical assumption that next-token probabilities follow a Zipf distribution, comparing how the hyperpa-mameter of each method influences the order of perplexity. This particular Zipfian assumption may not fully capture the complicated probability distributions encoded by modern language models.

An earlier work (Meister et al., 2020) attempted to explain heuristic methods such as beam search as an implicit regularization imposed on MAP. However, the design of regularization term seems to lack statistical motivations. They also suggest a qualitative relationship between the regularization term and the uniformity of surprisal of generated sequences, but the detailed mechanism is not well understood mathematically.

Overall, each of them motivates the design of heuristic decoding methods to a certain extent, but to the best of our knowledge, we are not aware of any comprehensive theoretical framework that establishes optimality results.

## 2.2 TEXT GENERATION AS DECISION MAKING

Another line of work, though not directly working on the theory of heuristic decoding schemes, views the problem of text generation as optimizing the policy of a decision-making agent working in an environment with or without adversary. This perspective resonates with our rationale behind the Decoding Game. For example, Jacob et al. (2024) proposed a game between a generation strategy and a text quality discriminator, and empirically demonstrated the advantage of the decoding strategy at the Nash equilibrium of this game in multiple tasks. Other recent works such as Snell et al. (2023); Kim et al. (2023); Mudgal et al. (2024) modeled next-token generation as (token-level) Markov decision processes, and applied reinforcement learning techniques for controlled decoding.

## 2.3 ROBUST OPTIMIZATION AND REGULARIZATION

Our framework also draws a connection to robust optimization (Ben-Tal et al., 2009), which aims to find solutions with stable performance under data uncertainty or perturbations. Given the adversarial nature of uncertainty, robust optimization is typically formulated as a minimax problem that seek the best response to the worst-case data realizations. Interestingly, while the sparsity brought by truncation sampling can be seen as regularization, it is known that regularization and robustness are strongly correlated in various machine learning problems (Derman et al., 2021; Shaham et al., 2018; Bertsimas et al., 2011), as solving an optimization problem with regularization is equivalent to solving its non-regularized robust counterpart. For instance, lasso and ridge optimization can be reformulated as robust optimization problems, where the data matrix is subject to different types of perturbations: ridge regression corresponds to Frobenius norm-bounded perturbations while lasso corresponds to column-wise $\ell_2$-norm bounded perturbations (Shaham et al., 2018). The theory developed in this paper also confirms such an equivalence between robustness and regularization.

# 3 FORMULATION

## 3.1 NOTATIONS

Throughout, we use boldface letters to represent vectors and vector-valued mappings, and use blackboard bold letters to represent probability measures and expectations. For a sequence $(x_0, x_1, \ldots, x_T)$, we define $x_{<t} = (x_0, \ldots, x_{t-1})$. For a vector $\boldsymbol{a} = (a_1, \ldots, a_d)$, $\boldsymbol{a}_{1:i} = (a_1, \ldots, a_i)$ is the vector extracting its first $i$ components. The $\ell_p$ norm ($p \geq 1$) of $\boldsymbol{a}$ is defined as $\|\boldsymbol{a}\|_p = (\sum_{i=1}^d |a_i|^p)^{1/p}$, with $\ell_\infty$ norm $\|\boldsymbol{a}\|_\infty = \max_{i \leq d} |a_i|$. The total variation (TV) distance between two probability vectors $\boldsymbol{p}, \boldsymbol{q}$ is defined as $d_{\text{TV}}(\boldsymbol{p}, \boldsymbol{q}) = \frac{1}{2} \|\boldsymbol{p} - \boldsymbol{q}\|_1$. For a function $f : \mathbb{R} \to \mathbb{R}$, $f(\boldsymbol{a})$ represents the elementwise application of $f$ to the vector $\boldsymbol{a}$, and $\boldsymbol{a}/\boldsymbol{b}$ represents

the elementwise division of $\boldsymbol{a}$ by $\boldsymbol{b}$. For a finite set $\mathcal{V}$, $\Delta(\mathcal{V})$ denotes the probability simplex of dimension $|\mathcal{V}|$, and $\mathcal{V}^t$ denotes the Cartesian product $\mathcal{V} \times ... \times \mathcal{V}$ ($t$ times). We always assume that optimization variables have to satisfy probability constraints (e.g., lying in the probability simplex, or the space of probability measures), and will not specify them explicitly for conciseness.

## 3.2 Decoding Game

We describe the $T$-step Decoding Game in full detail. Suppose $\mathcal{V} = \{1, 2, ..., d\}$ is the vocabulary of all $d$ tokens, and the natural language follows a true distribution $\mathbb{P}$. Upon training, we obtain a language model $\widehat{\mathbb{P}}$ that approximates the true $\mathbb{P}$. Beginning with a prescribed context $X_0 = (\text{prompt}, \langle \text{BOS} \rangle)$, we generate a sequence $(X_1, \ldots, X_T)$ with a possibly randomized decoding strategy, which is represented by another measure $\mathbb{Q}$ such that next tokens are selected with probability $\mathbb{Q}(X_t \mid X_{<t})$. We can then evaluate the quality of the generated sequence by testing whether the true distribution $\mathbb{P}$ is also likely to yield such a sequence. Specifically, for a typical sequence generated from $\mathbb{Q}$, we use its log-likelihood in $\mathbb{P}$-measure as the criterion

$$\mathcal{L}^T(\mathbb{Q}, \mathbb{P}) = \mathbb{E}_{\mathbb{Q}} \log \mathbb{P}(X_1, \ldots, X_T \mid X_0),$$

which is also the negative cross-entropy between $\mathbb{Q}$ and $\mathbb{P}$ when viewed as distributions on $\mathcal{V}^T$.

The $T$-step Decoding Game is a two-player zero-sum game on this criterion between Strategist (S) and Nature (N), where player S chooses $\mathbb{Q}$ (by choosing a decoding strategy) to maximize the criterion of the generation, while player N chooses $\mathbb{P}$ within a neighborhood of $\widehat{\mathbb{P}}$ to minimize it. This gives rise to the following formulation:

$$\max_{\mathbb{Q}} \min_{\mathbb{P} \in N(\widehat{\mathbb{P}})} \mathcal{L}^T(\mathbb{Q}, \mathbb{P}) = \max_{\mathbb{Q}} \min_{\mathbb{P} \in N(\widehat{\mathbb{P}})} \mathbb{E}_{\mathbb{Q}} \log \mathbb{P}(X_1, \ldots, X_T \mid X_0). \tag{MDG}$$

In other words, without knowing $\mathbb{P}$ specifically, player S seeks a strategy to optimize the objective $\mathcal{L}^T$ in the worst case among $N(\widehat{\mathbb{P}})$.

As a starting point of further understanding of the multi-step game, we build up a picture at $T = 1$. In this case, the probability measure $\mathbb{P}$ is represented by a $d$-dimensional probability vector $\boldsymbol{p} \in \Delta(\mathcal{V})$, where $p_i = \mathbb{P}(X_1 = i \mid X_0)$. Similarly, the one-step strategy $\mathbb{Q}$ is represented by $\boldsymbol{q} \in \Delta(\mathcal{V})$. We use TV distance to define the neighborhood $N(\hat{\boldsymbol{p}}) = \{\boldsymbol{p} : d_{\text{TV}}(\boldsymbol{p}, \hat{\boldsymbol{p}}) \leq \epsilon\}$, leading to the one-step Decoding Game

$$\max_{\mathbb{Q}} \min_{\mathbb{P} \in N(\widehat{\mathbb{P}})} \mathcal{L}^1(\mathbb{Q}, \mathbb{P}) = \max_{\boldsymbol{q}} \min_{\boldsymbol{p} \in N(\hat{\boldsymbol{p}})} \boldsymbol{q}^\top \log \boldsymbol{p}. \tag{ODG}$$

Our theoretical analysis in Section 4 will be mainly devoted to the one-step setting (ODG). Before that, we shall still take a deeper look into the general (MDG) and justify why the one-step game is a representative case that captures the essence of the general setting.

## 3.3 Reduction from multiple steps

In the multi-step setting, given a context $x_{<t} \in \mathcal{V}^{t-1}$, a probability measure $\mathbb{P}$ computes the conditional next-token distribution $\mathbb{P}(\cdot \mid x_{<t})$. Similar to one-step setting, such a next-token distribution corresponds to a $d$-dimensional probability vector $\boldsymbol{p}_t(x_{<t}) \in \Delta(\mathcal{V})$. We define the neighborhood in (MDG) as

$$N(\widehat{\mathbb{P}}) = \left\{ \mathbb{P} : d_{\text{TV}}(\boldsymbol{p}_t(x_{<t}), \hat{\boldsymbol{p}}_t(x_{<t})) \leq \epsilon, \ \forall x_{<t} \in \mathcal{V}^{t-1} \text{ and } t \leq T \right\},$$

which, as an extension from the one-step case, controls the dissimilarity between any pairs of conditional next-token distribution in TV distance.[1]

Then, we can recast (MDG) as

$$\max_{\mathbb{Q}} \min_{\mathbb{P} \in N(\widehat{\mathbb{P}})} \mathbb{E}_{\mathbb{Q}} \log \mathbb{P}(X_1, \ldots, X_T \mid X_0) = \max_{\mathbb{Q}} \min_{\mathbb{P} \in N(\widehat{\mathbb{P}})} \sum_{t=1}^{T} \mathbb{E}_{\mathbb{Q}} \log \mathbb{P}(X_t \mid X_{<t})$$

---

[1]It can also be interpreted as the $\epsilon$-ball of the TV-sup distance $d(\mathbb{P}, \widehat{\mathbb{P}}) = \max\{d_{\text{TV}}(\boldsymbol{p}_t(x_{<t}), \hat{\boldsymbol{p}}_t(x_{<t})) : x_{<t} \in V^{t-1}, t \leq T\}$, which is half of the $(1, \infty)$ mixed norm (Horn & Johnson, 1985) of the matrix concatenating all the conditional distributions difference $\boldsymbol{p}_t(x_{<t}) - \hat{\boldsymbol{p}}_t(x_{<t})$ as its columns.

$$
\begin{aligned}
&= \max_{\mathbb{Q}} \min_{\mathbb{P} \in N(\widehat{\mathbb{P}})} \sum_{t=1}^{T} \mathbb{E}_{X_{<t} \sim \mathbb{Q}} [\mathbb{E}_{X_t \sim \mathbb{Q}(\cdot | X_{<t})} [\log \mathbb{P}(X_t \mid X_{<t})]] \\
&= \max_{\mathbb{Q}} \min_{\mathbb{P} \in N(\widehat{\mathbb{P}})} \sum_{t=1}^{T} \mathbb{E}_{X_{<t} \sim \mathbb{Q}} [\boldsymbol{q}_t(X_{<t})^{\top} \log \boldsymbol{p}_t(X_{<t})] \\
&= \max_{\mathbb{Q}} \sum_{t=1}^{T} \mathbb{E}_{X_{<t} \sim \mathbb{Q}} \left[ \min_{\boldsymbol{p}_t(X_{<t}) \in N(\hat{\boldsymbol{p}}_t(X_{<t}))} \boldsymbol{q}_t(X_{<t})^{\top} \log \boldsymbol{p}_t(X_{<t}) \right].
\end{aligned} \tag{1}
$$

Here, (1) uses the fact that the minimization problem is inherently separable in each next-token distributions $\boldsymbol{p}_t(x_{<t})$.

Directly solving (MDG) or (1) is computationally intractable. Theoretically, one can exploit the recursive structure in (1) and apply dynamic programming, but the scale of the problem grows as $\Omega(d^T)$, and such an approach would take $\mathrm{poly}(d^T)$ time. Considering the computational cost of working on a modern LLM, in this paper we turn to local solutions that do not probe into the global structure of the problem. It is given by a locally optimal mechanism

$$
\tilde{\boldsymbol{q}}_t(x_{<t}) = \operatorname*{argmax}_{\boldsymbol{q}_t(x_{<t})} \min_{\boldsymbol{p}_t(x_{<t}) \in N(\hat{\boldsymbol{p}}_t(x_{<t}))} \boldsymbol{q}_t(x_{<t})^{\top} \log \boldsymbol{p}_t(x_{<t}) \quad \forall x_{<t} \in \mathcal{V}^{t-1}, \tag{2}
$$

thus going back to (ODG). In addition to significantly alleviating the computational cost, it turns out that such a local mechanism also provides the optimal worst-case performance over all decision processes that do not exploit future information of $\widehat{\mathbb{P}}$. We state this result formally next.

**Definition 3.1.** *We say a strategy $\mathbb{Q} = \mathbb{Q}(\widehat{\mathbb{P}})$ has no foresight if for any $t \leq T$, we have $\boldsymbol{q}_t(x_{<t}; \widehat{\mathbb{P}}) = \boldsymbol{q}_t(x_{<t}; \widehat{\mathbb{P}}')$ for any $\widehat{\mathbb{P}}$ and $\widehat{\mathbb{P}}'$ satisfying $\hat{\boldsymbol{p}}_s(x_{<s}) = \hat{\boldsymbol{p}}'_s(x_{<s}) \ \forall s \leq t$.*

**Assumption 3.2.** $\epsilon < \|\hat{\boldsymbol{p}}_t(x_{<t})\|_\infty$ *for all $t \leq T$ and $x_{<t} \in \mathcal{V}^{t-1}$.*

**Proposition 3.3.** *Given arbitrary $\widehat{\mathbb{P}}$ from the space of probability measures on $\mathcal{V}^T$, let $\mathbb{Q} = \mathbb{Q}(\widehat{\mathbb{P}})$ be any strategy with no foresight. Moreover, let $\mathbb{P}^* = \mathbb{P}^*(\widehat{\mathbb{P}}, \mathbb{Q})$ be the optimal strategy of player N against $\mathbb{Q}$. If Assumption 3.2 holds and $\widetilde{\mathbb{Q}} = \widetilde{\mathbb{Q}}(\widehat{\mathbb{P}})$ is the strategy induced by (2), then*

$$
\inf_{\widehat{\mathbb{P}}} \mathcal{L}^T(\widetilde{\mathbb{Q}}, \mathbb{P}^*) \geq \inf_{\widehat{\mathbb{P}}} \mathcal{L}^T(\mathbb{Q}, \mathbb{P}^*).
$$

Here, we make Assumption 3.2 so that the optimal value of the game is always well-defined, by staying away from $-\infty$. Proposition 3.3 provides worst-case justifications for optimizing the imminent one-step reward of the game at each timestep $t$. In what follows, we will be devoted to the one-step game (ODG) for theoretical analysis.

## 4 THEORETICAL ANALYSIS

In this section, we study the optimal strategies for both $\boldsymbol{p}$ and $\boldsymbol{q}$ in (ODG). We will show that the optimal $\boldsymbol{q}$ imposes an $\ell_\infty$-type regularization on the log-likelihood, and the optimal $\boldsymbol{p}$ solving the regularized maximization yields tail truncation-normalization sampling methods. Finally, we extend our analysis from log-likelihood to a general type of objectives, which recovers other heuristic methods and also highlights an exclusive advantage of log-likelihood.

For (ODG), we make the following assumptions.

**Assumption 4.1.** *The probabilities are strictly positive and, without loss of generality, sorted in decreasing order, i.e., $\hat{p}_1 \geq \hat{p}_2 \geq \cdots \geq \hat{p}_d > 0$.*

**Assumption 4.2.** *The distance budget $\epsilon$ satisfies $\hat{p}_d \leq \epsilon < \hat{p}_1$.*

### 4.1 $p$-STRATEGY: IMPLICIT REGULARIZATION

For a given $\boldsymbol{q}$, we have the following characterization for the optimal strategy of $\boldsymbol{p}$.

**Theorem 4.3.** *Under Assumption 4.1 and 4.2, let $\hat{\imath} = \max\{i : \hat{p}_i > \epsilon\}$. Define $\hat{\boldsymbol{w}} \in \mathbb{R}^{\hat{\imath}}$ elementwise by $\hat{w}_i = \frac{\epsilon}{\log \hat{p}_i/(\hat{p}_i - \epsilon)}$. Then, for a given $\boldsymbol{q}$, the optimal $\tilde{\boldsymbol{p}}$ for* (ODG) *satisfies*

$$\boldsymbol{q}^\top \log \tilde{\boldsymbol{p}} = \min_{\boldsymbol{p}:d_{TV}(\boldsymbol{p},\hat{\boldsymbol{p}})\leq \epsilon} \boldsymbol{q}^\top \log \boldsymbol{p} = \begin{cases} \boldsymbol{q}^\top \log \hat{\boldsymbol{p}} - \epsilon \left\| \boldsymbol{q}_{1:\hat{\imath}}/\hat{\boldsymbol{w}} \right\|_\infty, & \textit{if } q_i = 0, \ \forall i > \hat{\imath}; \\ -\infty, & \textit{otherwise.} \end{cases}$$

We give several remarks on Theorem 4.3.

**Tail truncation.** Due to the fact that $\lim_{x\downarrow 0} \log(x) = -\infty$, any $\boldsymbol{q}$ possessing a non-zero tail in the index set $\{i : \hat{p}_i \leq \epsilon\}$ will lead to a $-\infty$ objective value, because setting the corresponding $p_i$ to zero is within the reach of the adversary. This implies a hard truncation constraint $q_i = 0 \ \forall i \geq \hat{\imath}$, namely the optimal $\boldsymbol{q}$ must come without such a tail. However, more components can be additionally truncated in the maximization part, due to the $\ell_\infty$ regularization effect we derived.

**Tackling non-convexity.** The minimization problem in $\boldsymbol{p}$ is non-convex, because it has a concave objective. Therefore, it is inherently hard to characterize the structures of the optimal $\tilde{\boldsymbol{p}}$: it may not even be unique. However, under TV distance, the geometry of the feasible region is a polytope. This benefits our analysis because the minimum is known to be attained at the vertices (Horst, 1984), restricting the candidate solutions within a finite set. Switching to other dissimilarity metrics than TV distance, such as KL divergence, would change the geometry and require very different techniques, which is left for future work.

**Implicit regularization.** With optimal $\tilde{\boldsymbol{p}}$, the remaining part of the game is a regularized log-likelihood maximization in terms of $\boldsymbol{q}$:

$$\max_{\boldsymbol{q}} \boldsymbol{q}^\top \log \hat{\boldsymbol{p}} - \epsilon \left\| \boldsymbol{q}_{1:\hat{\imath}}/\hat{\boldsymbol{w}} \right\|_\infty, \quad \text{s.t. } q_i = 0, \ \forall i > \hat{\imath}$$

with an $\ell_\infty$-type regularization term $\left\| \boldsymbol{q}_{1:\hat{\imath}}/\hat{\boldsymbol{w}} \right\|_\infty$. Note that we have $\hat{\boldsymbol{w}} \approx \hat{\boldsymbol{p}}_{1:\hat{\imath}}$ by applying first-order approximation to the function $\log(1 + x)$. If no adversary is present ($\epsilon = 0$), there is no regularization effect and trivially, greedy sampling solves the one-step log-likelihood maximization. This establishes an equivalence between regularization and robustness against an adversary, which has been observed in the robust optimization literature; see Section 2.3.

## 4.2 $q$-STRATEGY: HEURISTIC SAMPLING METHODS

Now we present the optimal solution to the regularized maximization problem.

**Theorem 4.4.** *Under Assumption 4.1 and 4.2, let $\hat{\imath} = \max\{i : \hat{p}_i > \epsilon\}$, and $\hat{w}_i = \frac{\epsilon}{\log(\hat{p}_i/(\hat{p}_i - \epsilon))}$ for all $i \leq \hat{\imath}$. Define the threshold*

$$\hat{I} = \max \left\{ I : \sum_{i=1}^{I-1} \hat{w}_i \log(\hat{p}_i/\hat{p}_I) \leq \epsilon, \ \hat{p}_I > \epsilon \right\}.$$

*Then, the optimal $\tilde{\boldsymbol{q}}$ for* (ODG) *is given elementwise by*

$$\tilde{q}_i \propto \hat{w}_i \mathbb{1}_{(1 \leq i \leq \hat{I})}.$$

**Corollary 4.5.** *By first-order approximation, $\hat{w}_i \approx \hat{p}_i$, and hence $\tilde{q}_i \propto \hat{p}_i \mathbb{1}_{(1 \leq i \leq \hat{I})}$, where*

$$\hat{I} = \max \left\{ I : \sum_{i=1}^{I-1} \hat{p}_i \log(\hat{p}_i/\hat{p}_I) \leq \epsilon, \ \hat{p}_I > \epsilon \right\}.$$

*Thus, up to first-order approximation, a tail truncation-normalization sampling strategy is optimal.*

Clearly, Theorem 4.4 describes a sampling strategy that truncates tail probabilities and keeps only the subset $\{1, \ldots, \hat{I}\}$ as the support. Note that $\hat{I} \leq \hat{\imath}$ always holds. The new distribution on the support $\tilde{\boldsymbol{q}}_{1:\hat{I}}$ is *not* a simple rescaling of the original weights $\hat{\boldsymbol{p}}_{1:\hat{I}}$ by a normalization constant, which is different from past heuristic designs. However, according to Corollary 4.5, rescaling emerges as a first-order approximation to the optimal strategy.

Under first-order approximation, the truncation threshold of $\tilde{\boldsymbol{q}}$ has an information-theoretic interpretation. Note that if $I$ belongs to the support $\{1, \ldots, \hat{I}\}$, then $\sum_{i=1}^{I-1} \hat{p}_i \log(\hat{p}_i/\hat{p}_I) \leq \epsilon$, which is equivalent to

$$\log(1/\hat{p}_I) \leq H(\hat{\boldsymbol{p}}_{1:I-1}) + C_I,$$

where $H(\hat{\boldsymbol{p}}_{1:I-1})$ is the entropy of $\hat{\boldsymbol{p}}_{1:I-1}$, and $C_I := \log\left(1/\sum_{i=1}^{I-1}\hat{p}_i\right)+\epsilon/(\sum_{i=1}^{I-1}\hat{p}_i) \approx \epsilon$ for large $I$.[2] Thus, when constructing the support, for large $I$, we grow the existing support $\{1,\dots,I-1\}$ by adding $I$ if its self-information (surprisal) $\log(1/\hat{p}_I)$ is small compared to the existing entropy plus a small amount. In other words, this token selection mechanism modulates the total number of surprisals.

Notably, this truncation threshold is different from previous truncation-based methods like Nucleus sampling. However, one can still recover these strategies, say Nucleus sampling, by setting an appropriate $\epsilon$ that depends on $p$, the hyperparameter of Nucleus sampling, and also the distribution $\hat{p}_i$. This involves an adaptive $\epsilon$ that changes at every decoding step.

### 4.3 GENERALIZATION FROM LOG-LIKELIHOOD

Beyond the log-likelihood, our analysis can be extended to games with more general objectives, taking the form

$$\max_{\boldsymbol{q}} \min_{\boldsymbol{p}:d_{\mathrm{TV}}(\boldsymbol{p},\hat{\boldsymbol{p}})\leq\epsilon} \boldsymbol{q}^\top f(\boldsymbol{p}), \qquad\qquad (f\text{-ODG})$$

where $f : \mathbb{R} \to \mathbb{R}$ is applied elementwise on $\boldsymbol{p}$. We assume the following for ($f$-ODG).

**Assumption 4.6.** $f$ is non-decreasing and concave. Moreover, $(\epsilon, f)$ satisfies either of the following conditions:

(i) $\hat{p}_d \leq \epsilon < \hat{p}_1$, and $\lim_{x\downarrow 0} f(x) = -\infty$;

(ii) $0 < \epsilon < \hat{p}_d$, and $\sum_{i=1}^{d-1} \frac{f(\hat{p}_i)-f(\hat{p}_d+\epsilon)}{f(\hat{p}_i)-f(\hat{p}_i-\epsilon)} \geq 1$.

Clearly, our previous log-likelihood game (ODG), which uses $f(x) = \log(x)$, satisfies part (i) of this assumption. The following result establishes the solution to the general game ($f$-ODG), and encompasses Theorem 4.4 as a special case.

**Theorem 4.7.** *Under Assumption 4.1 and 4.6, let*

$$S_I = \sum_{i=1}^{I-1} \frac{f(\hat{p}_i)-f(\hat{p}_I)}{f(\hat{p}_i)-f(\hat{p}_i-\epsilon)},$$

*and define the threshold* $\hat{I} = \max\{I : S_I \leq 1,\ \hat{p}_I > \epsilon\}$. *Then, the optimal* $\tilde{\boldsymbol{q}}$ *for* ($f$-ODG) *is given elementwise by*

$$\tilde{q}_i \propto \frac{\epsilon}{f(\hat{p}_i)-f(\hat{p}_i-\epsilon)}\mathbb{1}_{(1\leq i\leq\hat{I})}. \qquad\qquad (3)$$

**Corollary 4.8.** *Suppose* $f$ *is also differentiable. By first-order approximation on* $f$, *we have* $\tilde{q}_i \propto \frac{1}{f'(\hat{p}_i)}\mathbb{1}_{(1\leq i\leq\hat{I})}$, *where*

$$\hat{I} = \max\left\{I : \sum_{i=1}^{I-1} \frac{f(\hat{p}_i)-f(\hat{p}_I)}{f'(\hat{p}_i)} \leq \epsilon,\ \hat{p}_I > \epsilon\right\}.$$

There are two interesting consequences of this general result.

**Exclusive advantage of log.** Log-likelihood is the *only* objective that makes it optimal to rescale the remaining probabilities by a normalization constant. This is because enforcing $\frac{1}{f'(x)} = x$ in Corollary 4.8 leads to $f(x) = \log(x)$ (up to a constant). Therefore, there is an exclusive connection between log-likelihood and rescaling.

**Temperature scaling.** Second, since rescaling is not necessarily optimal, it is worth understanding how we treat the remaining probabilities under another $f$. One example is $f(x) = \frac{x^{1-1/\tau}-1}{1-1/\tau}$, where $\tau \neq 1$.[3] Taking the derivative, we have $\frac{1}{f'(\hat{p}_i)} = \exp\left(\frac{\log\hat{p}_i}{\tau}\right) = \hat{p}_i^{1/\tau}$, hence recovering temperature sampling (Hinton, 2015) where the temperature is controlled by $\tau$. This shows the general game ($f$-ODG) is able to express a variety of sampling strategies.

---

[2]Note that $\frac{I-1}{\min\{d,\,1/\epsilon\}} \leq \sum_{i=1}^{I-1}\hat{p}_i \leq 1$ by Assumption 4.1 and the fact that $\hat{p}_i \geq \hat{p}_I > \epsilon$ for all $i < I$.

[3]Note that $\lim_{\tau\to 1} \frac{x^{1-1/\tau}-1}{1-1/\tau} = \log x$.

## 5 EXPERIMENTS

Building on the truncation and normalization mechanism given in the general theory, we propose *Game sampling* as outlined in Algorithm 1, and empirically evaluate its performance in text generation. Regarding the algorithm design, the objectives to our concern are $f(x) = \frac{x^{1-1/\tau}-1}{1-1/\tau}$ and $f(x) = \log(x)$, as a special case of $\tau = 1$. For better practical results, we relax the restrictions on the value of $\epsilon$ in Assumption 4.6.

---

**Algorithm 1** Game sampling

---

    **Input:** $0 < \epsilon \leq 1$, and $\tau > 0$
    **if** $\tau = 1$ **then**                                               ▷ log-likelihood objective
        compute $S_I = \sum_{i=1}^{I-1} \hat{p}_i \log(\hat{p}_i/\hat{p}_I)$
        find $\hat{I} = \max\{I : S_I \leq \epsilon\}$
        set $q_i = \hat{p}_i \mathbb{1}_{(1 \leq i \leq \hat{I})}$
    **else if** $\tau \neq 1$ **then**                                ▷ temperature sampling objective
        compute $S_I = (1 - 1/\tau)^{-1} \sum_{i=1}^{I-1} \hat{p}_i \left(1 - (\hat{p}_i/\hat{p}_I)^{1-1/\tau}\right)$
        find $\hat{I} = \max\{I : S_I \leq \epsilon\}$
        set $q_i = \hat{p}_i^{1/\tau} \mathbb{1}_{(1 \leq i \leq \hat{I})}$
    **end if**
    normalize $q_i$: $q_i \leftarrow q_i / \sum_{j=1}^{d} q_j$
    sample the next word based on the distribution $\boldsymbol{q}$

---

We conduct an open-ended text generation task using web text from the GPT-2 output dataset. For each of the 5,000 articles in the Webtext test set, we use the first 35 tokens as prompts, with a maximum generation length of 256 tokens. For each type of GPT-2 model (Small, Medium, Large, XL) (Radford et al., 2019), GPT-J-6B (Wang & Komatsuzaki, 2021), and Llama-2-7B (Touvron et al., 2023), we evaluate the following metrics:

1. **Perplexity**: The perplexity of the generated text under the corresponding model.
2. **Repetition frequency**: The fraction of generations with repetitions. A generation is considered repetitive if it contains at least two contiguous copies of the same phrase, of any length, at the token level.
3. **MAUVE score** (Pillutla et al., 2021): for comparison with human-written text, we use the corresponding human continuations from the test set, up to a maximum of 256 tokens.

A good performance is characterized by a high MAUVE score and close-to-human perplexity and repetition.

We compare seven decoding strategies: the proposed Game sampling (Algorithm 1), Nucleus sampling (Holtzman et al., 2020), Contrastive search (Su et al., 2022), Typical sampling (Meister et al., 2023), Basis-Aware sampling (BA-$\eta$) (Finlayson et al., 2024), Greedy sampling (using $\mathrm{argmax}_i \hat{p}_i$), and Pure sampling (sampling $i$ with probability $\hat{p}_i$). We use the best-performing hyperparameters for each strategy as determined by the MAUVE score.

The results are presented in Table 1, with a more detailed breakdown in Appendix B. Game sampling achieves the best performance in GPT-J-6B, GPT-2 XL, Medium, and Small models, and scores the second highest in GPT-2 Large and Llama-2-7B models, next to Nucleus sampling and BA-$\eta$ respectively. We remark that BA-$\eta$ involves matrix decomposition and operates at a higher computation cost compared to our method.

We also experiment with different values of $\epsilon$ and $\tau$, with details in Appendix B. In general, larger values of $\epsilon$ tend to produce better results in terms of higher MAUVE score, lower repetition frequency, and human-level perplexity. Smaller $\epsilon$ values reduce perplexity, but at the expense of more repetitions and lower MAUVE scores. In terms of $\tau$, the best performance for each $\epsilon$ choice was achieved at $\tau \approx 2$, yielding the highest MAUVE score. Similar to $\epsilon$, smaller $\tau$ values tend to reduce perplexity, but produce more repetitions with lower MAUVE scores. MAUVE score begins to decline when $\tau$ exceeds 2.

| Method | Perplexity | Repetition | MAUVE |
|---|---|---|---|
| Game ($\epsilon = 0.95, \tau = 2$) | **23.590** | **0.002** | **0.926** |
| Nucleus ($p = 0.9$) | 25.116 | 0.004 | 0.923 |
| Contrastive ($\alpha = 0.6$) | 2.267 | 0.835 | 0.035 |
| Typical ($p = 0.9$) | 10.785 | 0.019 | 0.888 |
| BA-$\eta$ ($\eta = 0.0001$) | 29.610 | **0.002** | 0.917 |
| Greedy | 2.163 | 0.886 | 0.0155 |
| Pure | 62.001 | 0.0002 | 0.845 |
| *Human* | *21.559* | *0.002* | *–* |

GPT-2 Small

| Method | Perplexity | Repetition | MAUVE |
|---|---|---|---|
| Game ($\epsilon = 0.95, \tau = 2$) | **17.499** | **0.002** | **0.945** |
| Nucleus ($p = 0.9$) | 19.221 | **0.002** | **0.945** |
| Contrastive ($\alpha = 0.6$) | 2.431 | 0.677 | 0.049 |
| Typical ($p = 0.9$) | 9.018 | 0.010 | 0.923 |
| BA-$\eta$ ($\eta = 0.0001$) | 24.119 | 0.0006 | 0.933 |
| Greedy | 2.247 | 0.808 | 0.029 |
| Pure | 48.553 | 0.0004 | 0.848 |
| *Human* | *15.923* | *0.002* | *–* |

GPT-2 Medium

| Method | Perplexity | Repetition | MAUVE |
|---|---|---|---|
| Game ($\epsilon = 0.99, \tau = 2.5$) | 15.458 | 0.001 | 0.947 |
| Nucleus ($p = 0.95$) | **13.699** | **0.002** | **0.954** |
| Contrastive ($\alpha = 0.6$) | 4.448 | 0.006 | 0.892 |
| Typical ($p = 0.9$) | 6.590 | 0.012 | 0.924 |
| BA-$\eta$ ($\eta = 0.0001$) | 15.223 | 0.0008 | **0.954** |
| Greedy | 2.169 | 0.760 | 0.039 |
| Pure | 21.952 | 0.0008 | 0.931 |
| *Human* | *13.755* | *0.002* | *–* |

GPT-2 Large

| Method | Perplexity | Repetition | MAUVE |
|---|---|---|---|
| Game ($\epsilon = 0.99, \tau = 2$) | **11.333** | **0.003** | **0.958** |
| Nucleus ($p = 0.95$) | 14.589 | **0.003** | 0.955 |
| Contrastive ($\alpha = 0.6$) | 5.235 | 0.006 | 0.912 |
| Typical ($p = 0.9$) | 7.342 | 0.011 | 0.932 |
| BA-$\eta$ ($\eta = 0.0001$) | 13.495 | **0.001** | 0.946 |
| Greedy | 2.411 | 0.672 | 0.065 |
| Pure | 23.505 | 0.0004 | 0.942 |
| *Human* | *12.319* | *0.002* | *–* |

GPT-2 XL

| Method | Perplexity | Repetition | MAUVE |
|---|---|---|---|
| Game ($\epsilon = 0.99, \tau = 2.5$) | 19.729 | **0.002** | **0.833** |
| Nucleus ($p = 0.99$) | 23.149 | **0.002** | 0.806 |
| Contrastive ($\alpha = 0.6$) | 6.573 | 0.007 | 0.715 |
| Typical ($p = 0.9$) | **8.747** | 0.016 | 0.769 |
| BA-$\eta$ ($\eta = 0.0001$) | 12.307 | **0.002** | 0.826 |
| Greedy | 2.603 | 0.731 | 0.043 |
| Pure | 27.327 | 0.001 | 0.798 |
| *Human* | *9.950* | *0.002* | *–* |

GPT-J-6B

| Method | Perplexity | Repetition | MAUVE |
|---|---|---|---|
| Game ($\epsilon = 0.95, \tau = 1.5$) | 14.000 | 0.128 | 0.858 |
| Nucleus ($p = 0.95$) | 31.125 | 0.154 | 0.853 |
| Contrastive ($\alpha = 0.6$) | **5.375** | 0.236 | 0.702 |
| Typical ($p = 0.9$) | 5.156 | 0.209 | 0.719 |
| BA-$\eta$ ($\eta = 0.0001$) | 8.375 | **0.002** | **0.890** |
| Greedy | 3.109 | 0.457 | 0.537 |
| Pure | 44.500 | 0.016 | 0.813 |
| *Human* | *6.469* | *0.002* | *–* |

Llama-2-7B

Table 1: Evaluations on open-ended text generation with different decoding strategies. Boldface values indicate the highest MAUVE score and the closest-to-human perplexity and repetition. The best-performing hyperparameters are selected for each strategy.

## 6 CONCLUSION

In this paper, we proposed Decoding Game, a two-player zero-sum game where a Strategist aims to maximize the log-likelihood of the generated text under the true probability measure, while an adversarial Nature seeks to distort the true measure within an $\epsilon$-error budget to degrade the text. After discussing the decomposibility of multi-step generation, we studied the optimal strategies for both players of the typical one-step Decoding Game. We proved that, as Nature enforces its optimal strategy, it imposes an $\ell_\infty$-type regularization on the log-likelihood maximization problem. By solving this regularized maximization in closed form, we identified tail truncation-normalization sampling as a first-order approximation to the optimal strategy.

We also generalized our theory from log-likelihood to a broader class of objectives. In deriving the general solution, we observed that log-likelihood is the only objective that makes it optimal to rescale the remaining probabilities by a normalizing constant. Selecting other types of objectives leads to different ways of treating the remaining probabilities, including temperature sampling. Moreover, we empirically evaluated the performance of Game sampling, a sampling strategy built upon the general theory, in open-ended text generation.

We believe that Decoding Game provides comprehensive theoretical understanding for the heuristic design of sampling strategies, by rigorously establishing regularization effect and optimality results. The statistically meaningful motivation and minimal assumptions behind Decoding Game open up its potential for future research, both theoretical and practical, on text generation strategies. For example, it would be interesting to generalize the metric beyond TV distance. Also, our formulation of the multi-step game shares similarities with token-level Markov decision processes, and efficiently tackling multi-step strategy via reinforcement learning would be another direction.

ACKNOWLEDGEMENTS

We extend our special thanks to the anonymous reviewers for their insightful comments and suggestions that greatly enhanced this work. We are also grateful to Danqi Chen for helpful discussions. Klusowski acknowledges financial support from the National Science Foundation through CAREER DMS-2239448.

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

# A  PROOFS

## A.1  PROOF OF PROPOSITION 3.3

For probability vectors $\boldsymbol{q}, \boldsymbol{p}, \hat{\boldsymbol{p}} \in \Delta(\mathcal{V})$, define $\mathcal{M}(\boldsymbol{q}, \hat{\boldsymbol{p}}) = \min_{\boldsymbol{p} \in N(\hat{\boldsymbol{p}})} \boldsymbol{q}^\top \log \boldsymbol{p}$, and $\overline{\mathcal{M}}(\hat{\boldsymbol{p}}) = \max_{\boldsymbol{q}} \min_{\boldsymbol{p} \in N(\hat{\boldsymbol{p}})} \boldsymbol{q}^\top \log \boldsymbol{p}$. Then, the $t$-step total rewards of no-foresight strategy $\mathbb{Q}(\widehat{\mathbb{P}})$ and locally optimal strategy $\widetilde{\mathbb{Q}}(\widehat{\mathbb{P}})$ are respectively given by

$$\mathcal{L}^t(\mathbb{Q}(\widehat{\mathbb{P}}), \mathbb{P}^*(\widehat{\mathbb{P}}, \mathbb{Q})) = \sum_{s=1}^{t} \mathbb{E}_{X_{<s} \sim \mathbb{Q}(\widehat{\mathbb{P}})}[\mathcal{M}(\boldsymbol{q}_s(X_{<s}), \hat{\boldsymbol{p}}_s(X_{<s}))] := \mathcal{R}^t(\mathbb{Q}(\widehat{\mathbb{P}}), \widehat{\mathbb{P}}),$$

$$\mathcal{L}^t(\widetilde{\mathbb{Q}}(\widehat{\mathbb{P}}), \mathbb{P}^*(\widehat{\mathbb{P}}, \widetilde{\mathbb{Q}})) = \sum_{s=1}^{t} \mathbb{E}_{X_{<s} \sim \widetilde{\mathbb{Q}}(\widehat{\mathbb{P}})}[\overline{\mathcal{M}}(\hat{\boldsymbol{p}}_s(X_{<s}))] := \widetilde{\mathcal{R}}^t(\widehat{\mathbb{P}}).$$

Since $\epsilon < \max_i \hat{p}_i$, $\overline{\mathcal{M}}(\hat{\boldsymbol{p}})$ is always bounded from below. Moreover, as the set-valued mapping $\hat{\boldsymbol{p}} \mapsto N(\hat{\boldsymbol{p}})$ satisfies upper and lower hemicontinuity and $N(\hat{\boldsymbol{p}})$ is compact, $\overline{\mathcal{M}}$ is continuous in $\hat{\boldsymbol{p}}$ by Berge's Maximum Theorem (Aliprantis & Border, 2006), which further implies the continuity of $\widetilde{\mathcal{R}}^t$. Since the space of $\widehat{\mathbb{P}}$ is compact, we conclude that infimum of $\widetilde{\mathcal{R}}^t$ can be attained at some $\widehat{\mathbb{P}}^*$, namely $\inf \widetilde{\mathcal{R}}^t(\widehat{\mathbb{P}}) = \widetilde{\mathcal{R}}^t(\widehat{\mathbb{P}}^*)$.

Now, if $\boldsymbol{q}_t(x_{<t}; \widehat{\mathbb{P}}^*) = \tilde{\boldsymbol{q}}_t(x_{<t}; \widehat{\mathbb{P}}^*) \ \forall t$, we are done. Otherwise, let $t_0$ be the first step such that $\boldsymbol{q}_{t_0}(x_{<t_0}; \widehat{\mathbb{P}}^*) \neq \tilde{\boldsymbol{q}}_{t_0}(x_{<t_0}; \widehat{\mathbb{P}}^*)$. We have

$$\sum_{s=1}^{t_0-1} \mathbb{E}_{X_{<s} \sim \mathbb{Q}(\widehat{\mathbb{P}}^*)}[\mathcal{M}(\boldsymbol{q}_s(X_{<s}), \hat{\boldsymbol{p}}_s^*(X_{<s}))] = \sum_{s=1}^{t_0-1} \mathbb{E}_{X_{<s} \sim \widetilde{\mathbb{Q}}(\widehat{\mathbb{P}}^*)}[\mathcal{M}(\tilde{\boldsymbol{q}}_s(X_{<s}), \hat{\boldsymbol{p}}_s^*(X_{<s}))],$$

$$\mathbb{E}_{X_{<t_0} \sim \mathbb{Q}(\widehat{\mathbb{P}}^*)}[\mathcal{M}(\boldsymbol{q}_{t_0}(X_{<t_0}), \hat{\boldsymbol{p}}_{t_0}^*(X_{<t_0}))] \leq \mathbb{E}_{X_{<t_0} \sim \widetilde{\mathbb{Q}}(\widehat{\mathbb{P}}^*)}[\mathcal{M}(\tilde{\boldsymbol{q}}_{t_0}(X_{<t_0}), \hat{\boldsymbol{p}}_{t_0}^*(X_{<t_0}))],$$

which implies $\mathcal{R}^{t_0}(\mathbb{Q}(\widehat{\mathbb{P}}^*), \widehat{\mathbb{P}}^*) \leq \widetilde{\mathcal{R}}^{t_0}(\widehat{\mathbb{P}}^*)$. Consider $\widehat{\mathbb{P}}^{**}$ defined as follows. For each $x_{<s} \in \mathcal{V}^{s-1}$,

$$\hat{\boldsymbol{p}}_s^{**}(x_{<s}) = \begin{cases} \hat{\boldsymbol{p}}_s^*(x_{<s}), & s \leq t_0, \\ \hat{\boldsymbol{p}}_s^*(x_{<s}^*) \text{ where } x_{<s}^* = \operatorname{argmin}_{x \in \mathcal{V}^{s-1}} \mathcal{M}(\tilde{\boldsymbol{q}}_s(x), \hat{\boldsymbol{p}}_s^*(x)), & s > t_0. \end{cases}$$

In words, $\widehat{\mathbb{P}}^{**}$ can be understood as shifting the future structure of $\widehat{\mathbb{P}}^*$ after $t_0$. Since the strategy $\mathbb{Q}(\widehat{\mathbb{P}})$ is defined to have no foresight, we have $\boldsymbol{q}_s(x_{<s}; \widehat{\mathbb{P}}^{**}) = \boldsymbol{q}_s(x_{<s}; \widehat{\mathbb{P}}^*)$ for $s \leq t_0$. Hence,

$$\mathcal{R}^{t_0}(\mathbb{Q}(\widehat{\mathbb{P}}^{**}), \widehat{\mathbb{P}}^{**}) \leq \widetilde{\mathcal{R}}^{t_0}(\widehat{\mathbb{P}}^*) \tag{4}$$

holds as well.

Due to our construction of $\widehat{\mathbb{P}}^{**}$, the future rewards after $t_0$ satisfy

$$\sum_{s=t_0+1}^{T} \mathbb{E}_{X_{<s} \sim \mathbb{Q}(\widehat{\mathbb{P}}^{**})}[\mathcal{M}(\boldsymbol{q}_s(X_{<s}), \hat{\boldsymbol{p}}_s^{**}(X_{<s}))] \leq \sum_{s=t_0+1}^{T} \max_{x_{<s} \in \mathcal{V}^{s-1}} \mathcal{M}(\boldsymbol{q}_s(x_{<s}), \hat{\boldsymbol{p}}_s^{**}(x_{<s}))$$

$$\leq \sum_{s=t_0+1}^{T} \max_{x_{<s} \in \mathcal{V}^{s-1}} \mathcal{M}(\tilde{\boldsymbol{q}}_s(x_{<s}), \hat{\boldsymbol{p}}_s^{**}(x_{<s}))$$

$$\leq \sum_{s=t_0+1}^{T} \mathbb{E}_{X_{<s} \sim \widetilde{\mathbb{Q}}(\widehat{\mathbb{P}}^*)}[\mathcal{M}(\tilde{\boldsymbol{q}}_s(X_{<s}), \hat{\boldsymbol{p}}_s^*(X_{<s}))],$$

namely

$$\mathcal{R}^T(\mathbb{Q}(\widehat{\mathbb{P}}^{**}), \widehat{\mathbb{P}}^{**}) - \mathcal{R}^{t_0}(\mathbb{Q}(\widehat{\mathbb{P}}^{**}), \widehat{\mathbb{P}}^{**}) \leq \widetilde{\mathcal{R}}^T(\widehat{\mathbb{P}}^*) - \widetilde{\mathcal{R}}^{t_0}(\widehat{\mathbb{P}}^*). \tag{5}$$

With (4) and (5), we conclude that

$$\inf_{\widehat{\mathbb{P}}} \mathcal{R}^T(\mathbb{Q}(\widehat{\mathbb{P}}), \widehat{\mathbb{P}}) \leq \mathcal{R}^T(\mathbb{Q}(\widehat{\mathbb{P}}^{**}), \widehat{\mathbb{P}}^{**}) \leq \widetilde{\mathcal{R}}^T(\widehat{\mathbb{P}}^*) = \inf_{\widehat{\mathbb{P}}} \widetilde{\mathcal{R}}^T(\widehat{\mathbb{P}}),$$

which proves the result.

## A.2 PROOF OF THEOREM 4.7

We shall only prove the general theorem, as Theorem 4.3 and 4.4 are direct consequences.

Consider the minimization problem

$$\min_{\boldsymbol{p} \in N(\hat{\boldsymbol{p}})} \boldsymbol{q}^\top f(\boldsymbol{p}), \tag{6}$$

where $N(\hat{\boldsymbol{p}}) = \{\boldsymbol{p} \in \Delta(\mathcal{V}) : d_{\text{TV}}(\boldsymbol{p}, \boldsymbol{q}) \leq \epsilon\}$.

The feasible region $N(\hat{\boldsymbol{p}})$ is a convex polytope since it is the intersection of two convex polytopes—the probability simplex $\Delta(\mathcal{V})$ and the $\epsilon$-TV-distance ball $\{\boldsymbol{p} : \frac{1}{2} \|\boldsymbol{p} - \hat{\boldsymbol{p}}\|_1 \leq \epsilon\}$. Moreover, due to concavity of $f$, it is easy to show that $\boldsymbol{q}^\top f(\boldsymbol{p})$ is concave in $\boldsymbol{p}$. It is well-known that minimizers of a concave function over a polytope are attained at one of the vertices (Horst, 1984). Now, we let $\mathcal{U}$ be the set of the vertices of $N(\hat{\boldsymbol{p}})$.

We will consider the two cases of the theorem separately, due to their differences in the geometry of the feasibility.

**Case 1**: $\epsilon < \hat{p}_d$, and $\sum_{i=1}^{d-1} \frac{f(\hat{p}_i) - f(\hat{p}_d + \epsilon)}{f(\hat{p}_i) - f(\hat{p}_i - \epsilon)} \geq 1$.

Since $\epsilon < \hat{p}_d$, the set $\mathcal{U}$ can be written as $\mathcal{U} = \{\hat{\boldsymbol{p}} - \epsilon \boldsymbol{e}_i + \epsilon \boldsymbol{e}_j : i \neq j\}$. Hence, we have

$$\min_{\boldsymbol{p} \in N(\hat{\boldsymbol{p}})} \boldsymbol{q}^\top f(\boldsymbol{p}) = \min_{\boldsymbol{p} \in \mathcal{U}} \boldsymbol{q}^\top f(\boldsymbol{p})$$

$$= \boldsymbol{q}^\top f(\hat{\boldsymbol{p}}) + \min_{i,j : i \neq j} \{q_i \left(f(\hat{p}_i - \epsilon) - f(\hat{p}_i)\right) + q_j \left(f(\hat{p}_j + \epsilon) - f(\hat{p}_j)\right)\}$$

$$= \boldsymbol{q}^\top f(\hat{\boldsymbol{p}}) - \max_{i,j : i \neq j} \{q_i g^-(\hat{p}_i) - q_j g^+(\hat{p}_i)\},$$

where $g^-(x) := f(x) - f(x - \epsilon)$, and $g^+(x) := f(x + \epsilon) - f(x)$. Taking this result into our game, the remaining $\boldsymbol{q}$-maximization part is equivalent to

$$\min_{\boldsymbol{q} \in \Delta(\mathcal{V})} \left[ -\boldsymbol{q}^\top f(\hat{\boldsymbol{p}}) + \max_{i,j : i \neq j} \{q_i g^-(\hat{p}_i) - q_j g^+(\hat{p}_i)\} \right]. \tag{7}$$

**Ordering of the optimal solution**. We claim that any optimal $\boldsymbol{q}^*$ has ordered elements, with $q_1^* \geq \cdots \geq q_d^*$. Observe that both $g^+$ and $g^-$ are non-increasing, since $f$ is a concave and non-decreasing function. Therefore, if a $\boldsymbol{q}$ has unordered elements, we can rearrange its elements it in descending order, and rearrangement inequality (Hardy et al., 1952) implies that that the term $-\boldsymbol{q}^\top f(\hat{\boldsymbol{p}})$ will decrease. Moreover, by reordering, the term $\max_{i,j : i \neq j} \{q_i g^-(\hat{p}_i) - q_j g^+(\hat{p}_i)\}$ will also decrease. This is because

$$\max_{i \neq j} \{q_i g^-(\hat{p}_i) - q_j g^+(\hat{p}_j)\} = \max_i \left\{ q_i g^-(\hat{p}_i) - \min_{j : j \neq i} q_j g^+(\hat{p}_j) \right\}$$

$$= \max_j \left\{ \max_{i : i \neq j} q_i g^-(\hat{p}_i) - q_j g^+(\hat{p}_j) \right\},$$

Thus, for any fixed $i$, if we reorder the rest of the elements, $\min_{j \neq i} q_j g^+(\hat{p}_j)$ will increase, making the entire term smaller. Further, by fixing $j$ and reordering by placing $q_i$ in the correct position, $\max_{i \neq j} q_i g^-(\hat{p}_i)$ will decrease. In total, rearranging $\boldsymbol{q}$ in descending order will decrease both terms, resulting in a lower overall objective.

**Analyzing KKT optimality**. Introducing dual variables $\boldsymbol{\lambda} \in \mathbb{R}_+^d, \nu \in \mathbb{R}$, the Lagrangian of (7) is given by

$$L(\boldsymbol{q}, \boldsymbol{\lambda}, \nu) := -\boldsymbol{q}^\top f(\hat{\boldsymbol{p}}) + \max_{i,j : i \neq j} \{q_i g^-(\hat{p}_i) - q_j g^+(\hat{p}_j)\} - \boldsymbol{\lambda}^\top \boldsymbol{q} + \nu \left( \sum_{i=1}^d q_i - 1 \right).$$

One can check that the objective in (7) is convex in $\boldsymbol{q}$. Moreover, since there exists $\tilde{\boldsymbol{q}} \in \text{relint}(\Delta(\mathcal{V}))$ with $\tilde{\boldsymbol{q}} > 0$, strong duality holds. Therefore, $\boldsymbol{q}^*$ is optimal if and only if there exists $\boldsymbol{\lambda}^*, \nu^*$ such that the following Karush-Kuhn-Tucker (KKT) conditions are satisfied (Boyd & Vandenberghe, 2004):

$$\boldsymbol{0} \in -f(\hat{\boldsymbol{p}}) + \partial \left( \max_{i,j : i \neq j} \{q_i^* g^-(\hat{p}_i) - q_j^* g^+(\hat{p}_j)\} \right) - \boldsymbol{\lambda}^* + \nu^* \boldsymbol{1}, \qquad \text{(first-order stationarity)}$$

$$\boldsymbol{q}^* \in \Delta(\mathcal{V}), \quad \boldsymbol{\lambda}^* \geq 0, \qquad\qquad\qquad \text{(primal-dual feasibility)}$$
$$\lambda_i^* q_i^* = 0 \quad \forall i, \qquad\qquad\qquad \text{(complementary slackness)}$$

where the subdifferential $\partial$ (Rockafellar, 1970) of the nonsmooth function inside represents the convex hull of the subgradients of the maximizing coordinates, given by

$$\partial\left(\max_{i \neq j}\left\{q_i^* g^-(\hat{p}_i) - q_j^* g^+(\hat{p}_j)\right\}\right) = \text{conv}\left(\mathcal{D}\right),$$

$$\mathcal{D} = \left\{g^-(\hat{p}_i)\boldsymbol{e}_i - g^+(\hat{p}_j)\boldsymbol{e}_j : i \neq j, \ q_i^* g^-(\hat{p}_i) - q_j^* g^+(\hat{p}_j) = \max_{i,j:i \neq j}\left\{q_i^* g^-(\hat{p}_i) - q_j^* g^+(\hat{p}_j)\right\}\right\}.$$

Now we show that $\boldsymbol{q}^*$ defined by $q_i^* = \frac{c}{g^-(\hat{p}_i)}\mathbb{1}_{(1 \leq i \leq I^*)}$ satisfies KKT conditions for some dual variables $\boldsymbol{\lambda}^*, \nu^*$, where $c$ is a normalizing constant. Let

$$\mathcal{J} := \{i : q_i^* g^-(\hat{p}_i) = c\} = \{1 \leq i \leq I^*\},$$
$$\mathcal{N} := \{i : q_i^* g^+(\hat{p}_i) = 0\} = \{I^* < i \leq d\}.$$

Then, as $S_I$ is non-decreasing in $I$, we have

$$\sum_{k=1}^{I^*-1} \frac{f(\hat{p}_k) - f(\hat{p}_i)}{g^-(\hat{p}_k)} \leq 1, \quad \forall i \in \mathcal{J}, \tag{8}$$

and

$$\sum_{k=1}^{I^*-1} \frac{f(\hat{p}_k) - f(\hat{p}_i)}{g^-(\hat{p}_k)} > 1, \quad \forall i \in \mathcal{N}. \tag{9}$$

Moreover, since

$$S_d = \sum_{k=1}^{d-1} \frac{f(\hat{p}_k) - f(\hat{p}_d)}{g^-(\hat{p}_k)} > \sum_{k=1}^{d-1} \frac{f(\hat{p}_k) - f(\hat{p}_d + \epsilon)}{g^-(\hat{p}_k)} \geq 1,$$

we know that $I^* < d$ must hold, and $\mathcal{N}$ is always non-empty.

To show that KKT conditions are satisfied, it is equivalent to prove that there exist $\nu^*, \boldsymbol{\lambda}^* \geq 0$ with $\lambda_i^* = 0$ for $i \in \mathcal{J}$, and coefficients $\gamma_{ij} \geq 0$ for $(i, j) \in \mathcal{J} \times \mathcal{N}$ with $\sum_{i \in \mathcal{J}} \sum_{j \in \mathcal{N}} \gamma_{ij} = 1$ such that

$$-f(\hat{p}_i) + g^-(\hat{p}_i)\left(\sum_{j \in \mathcal{N}} \gamma_{ij}\right)\mathbb{1}_{(i \in \mathcal{J})} - g^+(\hat{p}_i)\left(\sum_{j \in \mathcal{J}} \gamma_{ji}\right)\mathbb{1}_{(i \in \mathcal{N})} - \lambda_i^*\mathbb{1}_{(i \in \mathcal{N})} + \nu^* = 0,$$

which is equivalent to

$$-f(\hat{p}_i) + g^-(\hat{p}_i)\left(\sum_{j \in \mathcal{N}} \gamma_{ij}\right) + \nu^* = 0, \quad i \in \mathcal{J}, \tag{10}$$

$$-f(\hat{p}_i) - g^+(\hat{p}_i)\left(\sum_{j \in \mathcal{J}} \gamma_{ji}\right) + \nu^* = \lambda_i^* \geq 0, \quad i \in \mathcal{N}. \tag{11}$$

The above linear system is satisfied for

$$\nu^* = \left(\sum_{k \in \mathcal{J}} \frac{1}{g^-(\hat{p}_k)}\right)^{-1}\left(\sum_{k \in \mathcal{J}} \frac{f(\hat{p}_k)}{g^-(\hat{p}_k)} - 1\right),$$

$$\gamma_{ij} = \frac{f(\hat{p}_i) - \nu^*}{g^-(\hat{p}_i)}\mathbb{1}_{(j=d)},$$

$$\lambda_i^* = \left(-f(\hat{p}_i) - g^+(\hat{p}_d)\mathbb{1}_{(i=d)} + \nu^*\right)\mathbb{1}_{(i \in \mathcal{N})}.$$

Moreover, (8) and (9) respectively imply that $\gamma_{ij} \geq 0$ and $\lambda_i^* \geq 0$ for all $I^* < i < d$. We also have $\lambda_d^* \geq 0$ because

$$\sum_{k=1}^{d-1} \frac{f(\hat{p}_k) - f(\hat{p}_d) - g^+(\hat{p}_d)}{g^-(\hat{p}_k)} = \sum_{k=1}^{d-1} \frac{f(\hat{p}_k) - f(\hat{p}_d + \epsilon)}{g^-(\hat{p}_k)} \geq 1.$$

Therefore, the above choices of $\nu^*$, $\gamma_{ij}$, and $\lambda^*$ satisfy the linear system and all constraints. Thus, $(q^*, \lambda^*, \nu^*)$ satisfy the KKT conditions, and hence $q^*$ is the optimal solution to problem ($f$-ODG).

**Case 2**: $\hat{p}_d \leq \epsilon < \hat{p}_1$, and $\lim_{x \downarrow 0} f(x) = -\infty$.

Let $\mathcal{A} = \{i : \hat{p}_i \leq \epsilon\}$ and $\mathcal{Q} = \{q \in \Delta(\mathcal{V}) : q_i = 0 \ \forall i \in \mathcal{A}\}$. Suppose we use some strategy $q \notin \mathcal{Q}$, i.e., there is some $j \in \mathcal{A}$ such that $q_j \neq 0$. Since $\lim_{x \downarrow 0} f(x) = -\infty$, the adversary can always find $p = \hat{p} - \hat{p}_j e_j$ that makes the objective $-\infty$. Thus, an optimal strategy must come from $\mathcal{Q}$. Similar to Case 1, the $p$-minimization part can be written in terms of the vertex set $\mathcal{U}$ as follows:

$$\begin{aligned}
\min_{p \in N(\hat{p})} q^\top f(p) &= \min_{p \in \mathcal{U}} q^\top f(p) \\
&= \min_{p \in \mathcal{U}_{\mathcal{A}}} q^\top f(p) \\
&= q^\top f(\hat{p}) + \min_{(i,j) \in \mathcal{C}} \{q_i (f(\hat{p}_i - \epsilon) - f(\hat{p}_i)) + q_j (f(\hat{p}_j + \epsilon) - f(\hat{p}_j))\} \\
&= q^\top f(\hat{p}) - \max_{(i,j) \in \mathcal{C}} \{q_i g^-(\hat{p}_i) - q_j g^+(\hat{p}_i)\} \\
&= q^\top f(\hat{p}) - \max_{i \notin \mathcal{A}} q_i g^-(\hat{p}_i), \tag{12}
\end{aligned}$$

where $\mathcal{U}_{\mathcal{A}} = \{\hat{p} - \epsilon e_i + \epsilon e_j : i \neq j, i \notin \mathcal{A}\}$, and $\mathcal{C} = \{(i,j) : i \neq j, i \notin \mathcal{A}\}$. (12) follows because $q_j = 0$ for any $j \in \mathcal{A}$. Thus, the problem of interest is equivalent to

$$\min_{q \in \mathcal{Q}} \left[ -q^\top f(\hat{p}) + \max_{i \notin \mathcal{A}} q_i g^-(\hat{p}_i) \right].$$

In other words, we only need to solve $q^*$ from a lower-dimensional problem

$$\min_{q \in \Delta(\mathcal{V}_{\mathcal{A}})} \left[ -q^\top f(\hat{p}) + \max_i q_i g^-(\hat{p}_i) \right],$$

where $\mathcal{V}_{\mathcal{A}}$ is a truncated vocabulary with $|\mathcal{V}_{\mathcal{A}}| = d - |\mathcal{A}|$.

**Ordering of the optimal solution**. Similar to Case 1, an optimal $q^*$ is ordered with $q_1^* \geq \cdots \geq q_d^*$.

**Analyzing KKT optimality**. The Lagrangian can be similarly defined as

$$L(q, \lambda, \nu) := -q^\top f(\hat{p}) + \max_i q_i g^-(\hat{p}_i) - \lambda^\top q + \nu \left( \sum_{i=1}^{d-|\mathcal{A}|} q_i - 1 \right),$$

and strong duality holds as well. The KKT conditions are

$$\begin{aligned}
0 &\in -f(\hat{p}) + \partial \left( \max_i q_i^* g^-(\hat{p}_i) \right) - \lambda^* + \nu^* \mathbf{1}, && \text{(first-order stationarity)} \\
q^* &\in \Delta(\mathcal{V}_{\mathcal{A}}), \quad \lambda^* \geq 0, && \text{(primal-dual feasibility)} \\
\lambda_i^* q_i^* &= 0 \quad \forall i, && \text{(complementary slackness)}
\end{aligned}$$

where $\partial (\max_i q_i^* g^-(\hat{p}_i)) := \mathrm{conv}(\{g^-(\hat{p}_i) e_i : q_i^* g^-(\hat{p}_i) = \max_i q_i^* g^-(\hat{p}_i)\})$. Let

$$\mathcal{J} = \{i : q_i^* g^-(\hat{p}_i) = c\} = \{1 \leq i \leq I^*\}, \quad \mathcal{N} = \{i : q_i^* g^-(\hat{p}_i) = 0\} = \{I^* < i \leq d - |\mathcal{A}|\},$$

where $c := \max_i q_i^* g^-(\hat{p}_i)$. It is sufficient to show that there exist $\nu^*$, $\lambda^* \geq 0$ with $\lambda_i^* = 0$ for $i \in \mathcal{J}$, and coefficients $\gamma_i \geq 0$ for $i \in \mathcal{J}$ with $\sum_{i \in \mathcal{J}} \gamma_i = 1$, such that

$$-f(\hat{p}_i) + \gamma_i g^-(\hat{p}_i) \mathbb{1}_{\{i \in \mathcal{J}\}} - \lambda_i^* \mathbb{1}_{\{i \in \mathcal{N}\}} + \nu^* = 0.$$

This is achieved by setting

$$\nu^* = \left(\sum_{k \in \mathcal{J}} \frac{1}{g^-(\hat{p}_k)}\right)^{-1} \left(\sum_{k \in \mathcal{J}} \frac{f(\hat{p}_k)}{g^-(\hat{p}_k)} - 1\right),$$

$$\gamma_i = \frac{f(\hat{p}_i) - \nu^*}{g^-(\hat{p}_i)} \geq 0, \quad \text{for } i \in \mathcal{J},$$

$$\lambda_i^* = (\nu^* - f(\hat{p}_i)) \mathbb{1}_{(i \in \mathcal{N})} \geq 0.$$

Moreover, $\gamma_i \geq 0$ and $\lambda_i^* \geq 0$ follow from the fact that $S_I \leq 1 \ \forall I \in \mathcal{J}$ and $S_I > 1 \ \forall I \in \mathcal{N}$, respectively.

## B ADDITIONAL EXPERIMENTS

In Tables 2 and 3, we present additional experimental results obtained using various choices of $\epsilon$ and $\tau$ in Game sampling algorithm. These experiments provide further insights into the performance and sensitivity of the model under different parameter settings. We also explored different values of $\epsilon \in \{0.1, 0.3, 0.5, 0.8, 0.9\}$ alongside different $\tau$ values. However, since the best performance was consistently achieved with $\epsilon = 0.95$ or $\epsilon = 0.99$, we report only those values here to highlight the effect of changing $\tau$.

As part of this evaluation, we also analyzed the point at which probabilities are truncated and renormalized in Game sampling and Nucleus sampling for a randomly selected article from the WebText test set, using the GPT-2 XL model. The GPT-2 model has a total vocabulary size of 50,000 tokens, so truncating the probability distribution can significantly reduce the set of candidate words for the next token. Figures 1a and 1b illustrate how these sampling strategies truncate the probability distribution. Figure 1a shows the distribution for the next word when using only 1 token as context, along with the index where probabilities are truncated and set to zero. In contrast, Figure 1b presents the distribution for the next word when using the first 35 tokens as context, providing more information for the model to generate the next word. With more context, the model is expected to be more certain about the next word, and the figure highlights the corresponding truncation points. Notably, Game sampling truncates a substantial portion of the 50,000-token distribution and dynamically adjusts the cutoff point based on the shape of the distribution (see Algorithm 1).

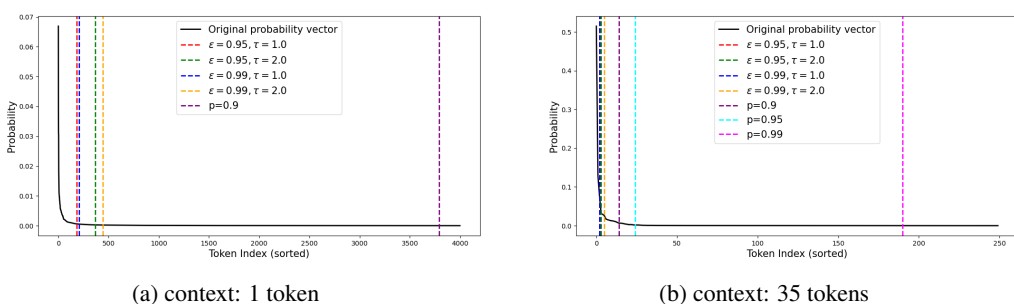

(a) context: 1 token                    (b) context: 35 tokens

Figure 1: Next-token probability distribution in GPT-2 XL model and truncation threshold of Game sampling and Nucleus sampling.

| $\epsilon$ | $\tau$ | Perplexity | Repetition | MAUVE |
|------|------|------------|------------|-------|
| 0.95 | 1.0 | 6.874 | 0.087 | 0.739 |
| 0.95 | 1.1 | 7.960 | 0.058 | 0.809 |
| 0.95 | 1.5 | 13.336 | 0.015 | 0.898 |
| 0.95 | 2.0 | 23.592 | 0.003 | 0.926 |
| 0.95 | 2.5 | 40.129 | 0.002 | 0.908 |
| 0.95 | 3.0 | 66.481 | 0.001 | 0.815 |
| 0.95 | 3.5 | 107.544 | 0.001 | 0.699 |
| 0.95 | 4.0 | 172.822 | 0.001 | 0.474 |
| 0.99 | 1.0 | 7.067 | 0.081 | 0.746 |
| 0.99 | 1.1 | 8.275 | 0.055 | 0.820 |
| 0.99 | 1.5 | 14.231 | 0.012 | 0.897 |
| 0.99 | 2.0 | 26.783 | 0.002 | 0.917 |
| 0.99 | 2.5 | 48.508 | 0.002 | 0.864 |
| 0.99 | 3.0 | 89.308 | 0.001 | 0.745 |
| 0.99 | 3.5 | 161.402 | 0.001 | 0.529 |
| 0.99 | 4.0 | 296.453 | 0.001 | 0.273 |

GPT-2 Small

| $\epsilon$ | $\tau$ | Perplexity | Repetition | MAUVE |
|------|------|------------|------------|-------|
| 0.95 | 1.0 | 6.067 | 0.048 | 0.858 |
| 0.95 | 1.1 | 6.804 | 0.037 | 0.883 |
| 0.95 | 1.5 | 10.423 | 0.010 | 0.926 |
| 0.95 | 2.0 | 17.499 | 0.003 | 0.945 |
| 0.95 | 2.5 | 28.738 | 0.001 | 0.919 |
| 0.95 | 3.0 | 46.973 | 0.001 | 0.858 |
| 0.95 | 3.5 | 78.152 | 0.001 | 0.721 |
| 0.95 | 4.0 | 132.77 | 0.001 | 0.475 |
| 0.99 | 1.0 | 6.176 | 0.047 | 0.845 |
| 0.99 | 1.1 | 6.947 | 0.033 | 0.879 |
| 0.99 | 1.5 | 11.019 | 0.008 | 0.941 |
| 0.99 | 2.0 | 19.482 | 0.002 | 0.938 |
| 0.99 | 2.5 | 34.662 | 0.002 | 0.911 |
| 0.99 | 3.0 | 63.555 | 0.001 | 0.792 |
| 0.99 | 3.5 | 120.889 | 0 | 0.497 |
| 0.99 | 4.0 | 243.844 | 0 | 0.257 |

GPT-2 Medium

| $\epsilon$ | $\tau$ | Perplexity | Repetition | MAUVE |
|------|------|------------|------------|-------|
| 0.95 | 1.0 | 4.596 | 0.066 | 0.823 |
| 0.95 | 1.1 | 4.972 | 0.050 | 0.856 |
| 0.95 | 1.5 | 6.851 | 0.013 | 0.909 |
| 0.95 | 2.0 | 9.883 | 0.005 | 0.942 |
| 0.95 | 2.5 | 14.084 | 0.002 | 0.942 |
| 0.95 | 3.0 | 19.634 | 0.002 | 0.930 |
| 0.95 | 3.5 | 27.779 | 0.001 | 0.913 |
| 0.95 | 4.0 | 39.256 | 0.001 | 0.837 |
| 0.99 | 1.0 | 4.683 | 0.066 | 0.826 |
| 0.99 | 1.1 | 5.083 | 0.046 | 0.861 |
| 0.99 | 1.5 | 7.130 | 0.010 | 0.917 |
| 0.99 | 2.0 | 10.629 | 0.006 | 0.947 |
| 0.99 | 2.5 | 15.958 | 0.001 | 0.947 |
| 0.99 | 3.0 | 24.128 | 0.001 | 0.919 |
| 0.99 | 3.5 | 37.613 | 0.001 | 0.845 |
| 0.99 | 4.0 | 60.031 | 0.001 | 0.685 |

GPT-2 Large

| $\epsilon$ | $\tau$ | Perplexity | Repetition | MAUVE |
|------|------|------------|------------|-------|
| 0.95 | 1.0 | 5.146 | 0.050 | 0.861 |
| 0.95 | 1.1 | 5.559 | 0.033 | 0.891 |
| 0.95 | 1.5 | 7.475 | 0.014 | 0.935 |
| 0.95 | 2.0 | 10.541 | 0.004 | 0.950 |
| 0.95 | 2.5 | 14.636 | 0.002 | 0.948 |
| 0.95 | 3.0 | 20.458 | 0.002 | 0.929 |
| 0.95 | 3.5 | 28.410 | 0.001 | 0.919 |
| 0.95 | 4.0 | 39.374 | 0.001 | 0.873 |
| 0.99 | 1.0 | 5.219 | 0.044 | 0.852 |
| 0.99 | 1.1 | 5.660 | 0.032 | 0.886 |
| 0.99 | 1.5 | 7.784 | 0.010 | 0.943 |
| 0.99 | 2.0 | 11.333 | 0.003 | 0.958 |
| 0.99 | 2.5 | 16.690 | 0.003 | 0.952 |
| 0.99 | 3.0 | 24.796 | 0.002 | 0.924 |
| 0.99 | 3.5 | 38.056 | 0.001 | 0.885 |
| 0.99 | 4.0 | 60.236 | 0.001 | 0.739 |

GPT-2 XL

Table 2: Evaluations on the text generated by different types of GPT-2 models using Game sampling under different hyperparameters.

| $\epsilon$ | $\tau$ | Perplexity | Repetition | MAUVE | $\epsilon$ | $\tau$ | Perplexity | Repetition | MAUVE |
|---|---|---|---|---|---|---|---|---|---|
| 0.95 | 1.0 | 5.757 | 0.069 | 0.640 | 0.95 | 1.0 | 8.500 | 0.131 | 0.842 |
| 0.95 | 1.1 | 6.285 | 0.049 | 0.670 | 0.95 | 1.1 | 9.938 | 0.134 | 0.831 |
| 0.95 | 1.5 | 8.528 | 0.015 | 0.759 | 0.95 | 1.5 | 14.000 | 0.128 | 0.858 |
| 0.95 | 2.0 | 12.313 | 0.005 | 0.794 | 0.95 | 2.0 | 23.875 | 0.149 | 0.843 |
| 0.95 | 2.5 | 17.210 | 0.003 | 0.811 | 0.95 | 2.5 | 36.250 | 0.162 | 0.834 |
| 0.95 | 3.0 | 24.362 | 0.001 | 0.801 | 0.95 | 3.0 | 52.000 | 0.173 | 0.813 |
| 0.95 | 3.5 | 33.905 | 0.002 | 0.778 | 0.95 | 3.5 | 63.750 | 0.174 | 0.797 |
| 0.95 | 4.0 | 48.921 | 0.001 | 0.664 | 0.95 | 4.0 | 87.000 | 0.182 | 0.753 |
| 0.99 | 1.0 | 5.897 | 0.066 | 0.664 | 0.99 | 1.0 | 8.938 | 0.130 | 0.831 |
| 0.99 | 1.1 | 6.436 | 0.046 | 0.687 | 0.99 | 1.1 | 10.250 | 0.134 | 0.845 |
| 0.99 | 1.5 | 8.957 | 0.013 | 0.762 | 0.99 | 1.5 | 15.625 | 0.136 | 0.854 |
| 0.99 | 2.0 | 13.263 | 0.004 | 0.809 | 0.99 | 2.0 | 26.625 | 0.153 | 0.840 |
| 0.99 | 2.5 | 19.729 | 0.002 | 0.833 | 0.99 | 2.5 | 41.750 | 0.165 | 0.822 |
| 0.99 | 3.0 | 29.696 | 0.002 | 0.791 | 0.99 | 3.0 | 60.000 | 0.181 | 0.806 |
| 0.99 | 3.5 | 46.506 | 0 | 0.720 | 0.99 | 3.5 | 84.500 | 0.178 | 0.759 |
| 0.99 | 4.0 | 77.289 | 0.001 | 0.522 | 0.99 | 4.0 | 119.500 | 0.177 | 0.686 |

GPT-J-6B                                     Llama-2-7B

Table 3: Evaluations on the text generated by GPT-J-6B and Llama-2-7B models using Game sampling under different hyperparameters.

