# OpenReview forum: "Decoding Game: On Minimax Optimality of Heuristic Text Generation Strategies"
_ICLR.cc/2025/Conference — ICLR 2025 Poster_

### Official Review · Reviewer_d8ex · 2024-10-19

**Soundness:** 3
**Presentation:** 3
**Contribution:** 3
**Rating:** 8
**Confidence:** 3

**Summary:**

The paper derives stochastic decoding methods as the optimal strategy in an adversarial two-player game. Due to tractability issues, a one-step version of this game and a first order approximation for the resulting strategy is used. An explicit expression for the resulting regularization term added to the maximum likelihood objective is derived.

**Strengths:**

* It is valuable to have a proper theoretical derivation of heuristic decoding methods that have long been around in practice, and the paper provides this.
* the paper is well-written and neat
* the connection to robust optimization is interesting

**Weaknesses:**

* The empirical comparison is not state-of-the-art. For example, there are newer variants of greedy designed to overcome issues with repetitiveness, see contrastive search (see: https://huggingface.co/blog/introducing-csearch).
* The introduction sounds a bit like stochastic sampling strategies would always be preferable over deterministic optimization strategies like greedy search or beam search in practice. While there definitely is evidence that they perform better in some settings as given by the citations in the paper, there is also counter-evidence in other settings, e.g. [1], in particular in closed-end tasks. (Btw, the experiments in the paper are also limited to open-end text generation.) I understand that the authors preliminarily want to motivate stochastic decoding strategies in this paper and I do not want to diminish the motivation for them, but I still think it is important to at least acknowledge that the situation is more nuanced. For a recent empirical comparison, also see [2].

[1] "On Decoding Strategies for Neural Text Generators" [Wiher et al, 2022]

[2] "A Thorough Examination of Decoding Methods in the Era of LLMs" [Shi et al., 2024]

**Questions:**

See weaknesses.

**Details Of Ethics Concerns:**

-

---

> ### Author Response · Authors · 2024-11-20
>
> We thank the reviewer very much for their feedback and suggestions, especially on the nuances of the performance of different decoding strategies, both stochastic and deterministic, across different tasks. We address each point below.
>
> **Empirical comparison with state-of-the-art variants of greedy decoding:** We agree that it is important to compare newer searching methods. As suggested by the reviewer, we have conducted open-ended experiments on Constrastive Search and summarize the results here.
>
> In GPT-2 Small:
>
> | method | perplexity | repetition | MAUVE |
> | --- | --- | --- | --- |
> | Contrastive | 2.267 | 0.835 | 0.035 |
> | Greedy | 2.163 | 0.886 | 0.0155 |
> | Game | 23.590 | 0.002 | 0.926 |
> | Human | 21.559 | 0.002 | --- |
>
> In GPT-2 Medium:
>
> | method | perplexity | repetition | MAUVE |
> | --- | --- | --- | --- |
> | Contrastive | 2.431 | 0.677 | 0.049 |
> | Greedy | 2.247 | 0.808 | 0.0288 |
> | Game | 17.499 | 0.002 | 0.945 |
> | Human | 15.923 | 0.002 | --- |
>
> In GPT-2 Large:
>
> | method | perplexity | repetition | MAUVE |
> | --- | --- | --- | --- |
> | Contrastive | 4.448 | 0.0064 | 0.892 |
> | Greedy | 2.169 | 0.760 | 0.039 |
> | Game | 15.458 | 0.001 | 0.947 |
> | Human | 13.755 | 0.002 | --- |
>
> In GPT-2 XL:
>
> | method | perplexity | repetition | MAUVE |
> | --- | --- | --- | --- |
> | Contrastive | 5.235 | 0.0056 | 0.912 |
> | Greedy | 2.411 | 0.672 | 0.065 |
> | Game | 11.333 | 0.003 | 0.958 |
> | Human | 12.319 | 0.002 | --- |
>
> We also experimented on larger models as suggested by other reviewers. In GPT-J-6B:
>
> | method | perplexity | repetition | MAUVE |
> | --- | --- | --- | --- |
> | Contrastive | 6.573 | 0.007 | 0.715 |
> | Greedy | 2.603 | 0.731 | 0.043 |
> | Game | 19.729 | 0.002 | 0.833 |
> | Human | 9.95 | 0.002 | --- |
>
> And we are now evaluating Llama-2-7B models and will update you once it is ready.
>
> These results complement our current Table 1. We observe that Contrastive Search does significantly reduce repetition and improve MAUVE score, especially in the larger models. However, in these open-ended generation tasks, our method still maintains an edge. We believe this corresponds to the reviewer's comments that deterministic strategies tend to perform better in closed-end tasks.
>
> **Expanding the scope in introduction:** We thank the reviewer for suggesting a more comprehensive summary of the empirical performances of decoding strategies. We have revised our manuscript to present a more nuanced analysis of decoding strategies across different tasks, incorporating recent empirical studies including [Wiher et al., 2022] and [Shi et al., 2024]. Following the reviewer's comments, our expanded literature review now better acknowledges task-specific performance variations between stochastic and deterministic methods.
>
> Thank you again for your comprehensive and helpful review. Based on your feedback, we are now working on a revised version and will update you before the end of discussion period. During this time, please also let us know if there are any further questions.

---

> > ### Comment · Reviewer_d8ex · 2024-11-21
> >
> > Thank you for the reply. My concerns are sufficiently addressed. I will increase my score from 6 to 8.

---

> > > ### Author Response · Authors · 2024-11-22
> > >
> > > We are very grateful for your support and help with our submission. We would appreciate it very much if the rating score in your original review could be updated.

---

### Official Review · Reviewer_ivfP · 2024-10-28

**Soundness:** 3
**Presentation:** 3
**Contribution:** 3
**Rating:** 5
**Confidence:** 3

**Summary:**

The paper explores the gap between theoretical and practical decoding strategies in text generation for language models.The authors introduce a new framework, Decoding Game, which models text generation as a two-player game. They derive the optimal one-step strategy, showing adversarial regularization on likelihood maximization, with popular methods as first-order approximations. The framework also generalizes to include strategies like greedy search and temperature scaling. Numerical experiments support their theoretical findings.

**Strengths:**

- The analysis of the decoding process of LLMs is thoroughly considered.
- The proposed decoding is backed up theoretically
- The experimental results show somewhat better performance than baselines.

**Weaknesses:**

- What is the motivation for choosing the TV-sup norm to compute the difference between the true distribution and the approximated one by LLMs? Is there any theoretical justification behind this selection? If not, at least an ablation study showing the effectiveness of this norm is necessary.
- It is still ambiguous why the authors define the distance between the two mappings by the way around Line 248.
- The result in the Equation (1) is non-trivial. Detailed transformation of it is necessary. At first glance, I questioned the equivalence of the two sides.

**Questions:**

N/A

---

> ### Author Response · Authors · 2024-11-18
>
> We thank the reviewer very much for their suggestions and questions, especially on the norm and neighborhood notation in this paper. We address each point below.
>
> **Understanding TV-sup norm:** We define the TV-sup norm as the maximum total variation distance between pairs of conditional next-token distributions. Formally, $||\mathbb P-\widehat{\mathbb P}||\_{\text{TV},\infty}\leq\epsilon$ means that for all contexts $x\_{<t}$, the conditional next-token distributions has at most $\epsilon$ TV distance: $d_\text{TV}(\mathbb P(\cdot|x_{<t}),\widehat{\mathbb P}(\cdot|x_{<t}))\leq\epsilon$.
>
> We use this particular definition because it naturally extends the one-step game constraint $d_\text{TV}(p,\hat p)\leq\epsilon$ to the multi-step setting. Since our goal is to reduce the multi-step setting back to the one-step setting, such a choice of norm is the most convenient.
>
> Equivalently, if we arrange all conditional next-token distributions as columns of a matrix, our TV-sup norm corresponds to the maximum of the $\ell_1$-norm (by definition, twice as the TV norm) of each column, also known as the $(1,\infty)$ mixed norm:
>
> $||A||\_{1,\infty}=\max_j\sum_{i=1}^{d}|a_{ij}|$.
>
> See, for instance, Example 5.6.4 of [1], a standard textbook in matrix analysis.
>
> This norm finds broad applications in probability and statistics literature. For example, Chapter 3 of [2] on asymptotics of Markov processes, and [3,4] on regression and signal processing.
>
> We acknowledge that we need to improve the readability of our current presentation on TV-sup norm. Based on your feedback, we have revised our presentation to better emphasize how the neighborhood $N(\widehat{\mathbb P})$ builds upon the one-step TV norm and facilitates the reduction to one-step games, and eased the notations involved.
>
> **Understanding the direct products in Equation (1):** Let us explain how the $(1,\infty)$ mixed norm naturally leads to a direct product decomposition. Consider a matrix $A=[\mathbf a_1,\mathbf a_2,...,\mathbf a_N]\in\mathbb R^{d\times N}$. The following equivalences hold:
>
> $||A||\_{1,\infty}\leq\epsilon \Leftrightarrow \max_i\{||\mathbf a_i||_1}\leq\epsilon \Leftrightarrow ||\mathbf a_i||_1\leq\epsilon\ \forall i$.
>
> This yields the decomposition:
>
> {$A:||A||_{1,\infty}\leq\epsilon$} $=$ {$\mathbf a_1:||\mathbf a_1||_1\leq\epsilon$} $\times...\times$ {$\mathbf a_N:||\mathbf a_N||_1\leq\epsilon$}.
>
> Equation (1) follows from this fundamental principle: a uniform bound on the maximum implies the same bound holds individually for each component.
>
> Thank you again for your comprehensive and helpful review. Based on your feedback, we are now working on a revised version and will update you before the end of discussion period. During this time, please also let us know if there are any further questions.
>
> **References**
>
> [1] R. A. Horn and C. R. Johnson, *Matrix Analysis*. Cambridge: Cambridge University Press, 1985.
>
> [2] Daniel W. Stroock. *An introduction to Markov processes*. Vol. 230. Springer Science & Business Media, 2013.
>
> [3] Matthieu Kowalski. "Sparse regression using mixed norms." *Applied and Computational Harmonic Analysis* 27.3 (2009): 303-324.
>
> [4] Dongmin Kim, Suvrit Sra, and Inderjit Dhillon. "A scalable trust-region algorithm with application to mixed-norm regression." *27th International Conference on Machine Learning (ICML)*, 2010.

---

> > ### Comment · Reviewer_ivfP · 2024-11-23
> >
> > Thanks for the author's' responses. I tend to keep my current score.

---

### Official Review · Reviewer_1fhp · 2024-11-04

**Soundness:** 4
**Presentation:** 3
**Contribution:** 3
**Rating:** 8
**Confidence:** 4

**Summary:**

The paper proposes a theoretical framework that captures various decoding strategies used in practice, as well as provides a generalized form of such a technique and some experiments to support it.

**Strengths:**

The theoretical framework is well justified and rigorously supported by proofs
Statements are accurate and notation is consistent. Assumptions are clearly stated and natural.
The problem that is addressed is practically significant.
The framework generalizes previous decoding schemes opening the door to new, theoretically-grounded, heuristics.

**Weaknesses:**

In Prop 3.1, the authors could mention why we need the $\epsilon < max \hat{p}$ - that if we assign non-zero measure to $x_{<t}$, we’d get a cost of $-\infty$.
Theorem 4.3 doesn’t say anything about how the $p$ that yields the optimal solution looks like, nor what role $\hat{w}$ plays (in the min problem; the max one is clear). Does that lack structure or could it be added to the Theorem statement?
While the game approach to it is intuitive to some, it might be worth emphasizing the perspective of optimizing the worst case scenario: there are a couple of places where you phrase the framework, but don’t necessarily argue why nature “should” be adversarial. Optimizing the worst case scenario makes it more clear to people that are not used to thinking about games (since after all you just choose the approach that has the best guarantees under the TV assumption).
It’d be helpful to present the more general case secondary to what you get when you use a greedy local approach (which is done in section 3.4 and then used throughout). Once you decide to do local optimization rather than global, you can jump straight to Equation (4) and while Section 3.3 is well-written and objectively accurate, it is the hardest part to read of the whole paper and people may lose the important take-aways because of getting confused there.

Minor:
Remark at line 339 seems to be more relevant right before Corollary 4.5.
Line 54-55 repeats “From a statistical perspective”
It is worth making explicit that X_0 is a multidimensional object (as opposed to X_{t>0})
Line 243: $N (\hat{P})$ to be $\hat{P}$.. - an extra $\hat{P}$.

**Questions:**

“Depending on the desired effect, the truncation threshold or S may be determined from various quantities like the truncated vocabulary size, truncated probability weight, entropy, and so on. ” - this is confusing since the truncated vocabulary size, probability weight etc are themselves a function of S. What did you mean here?
Regarding the experiments, have you tried any of the methods that drop non-tail elements? (such as truncation sampling by Finlayson et al. (2024)). It would be interesting to see how a different-pattern sampling compares.
Is there any reason in the remark around line 376 for $C_{I-1}$ to not dominate the entropy? It’s not obvious how to interpret that.
Can your method be adapted to deal with other sorts of measure distances? The most natural one would be bounded cross entropy since this is what training usually optimizes

---

> ### Author Response · Authors · 2024-11-18
>
> **[1/2]**
>
> Thank you for your thorough review and insightful questions. Your feedback has helped us significantly improve the paper, particularly its presentation. We address each of your points below.
>
> ## In Weaknesses section,
>
> **Why we need $\epsilon<\max\hat p$ in Proposition 3.1:** We agree that this constraint is crucial, as it prevents the optimal value of the game from becoming $-\infty$. We have added an explanation to this in the revised version.
>
> **The structure of optimal $p$ and the role of $\hat w$ in Theorem 4.3:** We have added a characterization of the optimal $p$. The core minimization problem, $\min_{p:d_\text{TV}(p,\hat p)\leq\epsilon}q^\top\log p$, involves minimizing a concave function over the polytope defined by the $\epsilon$-ball in total variation distance.
>
> Due to non-convexity, the optimal $p$ may not be unique, and its structure is not trivial. However, it is known that such a problem always attains the minimum at a vertex of the polytope. Specifically, the optimal $p$ belongs to the set {$\hat p-\epsilon e_i+\epsilon e_j:i\neq j$}, where $e_i$ denotes the $i$-th standard basis vector in $\mathbb R^d$. Each candidate solution has the form $(\hat p_1,...,\hat p_i-\epsilon,...,\hat p_j+\epsilon,...,\hat p_d)$.
>
> The optimal $p$ has a more complicated structure than the optimal $q$, because we minimize over a non-convex problem with a concave objective, while maximizing over a convex one. However, our polytope characterization simplifies this analysis considerably. The optimization reduces to selecting the optimal $p$ from a finite set of candidates, expressed as:
>
> $$\min_{i\leq\hat\iota}q_{i}(\log (p_i-\epsilon)-\log p_i),$$
>
> which, in vector notation, becomes
>
> $$\min_{i\leq\hat\iota}q_i(\log (p_i-\epsilon)-\log p_i)=-||q_{1:\hat\iota}/\hat w||_\infty.$$
>
> This explains both the role of $\hat w$ and the emergence of the $\ell_\infty$ norm: $\hat w$ effectively vectorizes the vertex selection process, and by doing so we can explicitly see the regularization term. We have added this geometric intuition to the main text to complement the formal proofs.
>
> **Interpreting the game as worst-case optimization:** Optimizing the worst-case scenario (robust optimization) is definitely an important perspective supplementing our game-theoretic approach. We have strengthened our analysis by explicitly connecting the game-theoretic and robust optimization perspectives.
>
> **Clearer presentation of the general case:** Following your suggestion, we have reorganized the paper to improve its pedagogical flow. The new Section 3.2 introduces the one-step game. Section 3.3 then extends these ideas to the multi-step case, with simplified notation to enhance readability. We hope this progression from specific to general cases makes the theoretical development more accessible.

---

> ### Author Response · Authors · 2024-11-18
>
> **[2/2]**
>
> ## In Questions section,
>
> **What do we mean by the sentence about truncation threshold:** We wanted to express the fact that different decoding methods have different mechanisms for determining $S$. For example, in top-k sampling, $S$ is simply {$1,...,k$}, where $k$ is a fixed size parameter. In nucleus sampling, $S$ is {$1,...,i^*$} where $i^*=\max${$i:\sum_{j=1}^i\hat p_j\leq p$}, with $p$ representing the desired cumulative probability mass.
>
> We have rephrased this sentence to improve clarity: "Here, different designs define the truncation set $\mathcal S$ using distinct criteria: top-$k$ sampling uses a fixed size threshold, Nucleus sampling uses cumulative probability mass, and entropy-based methods use information-theoretic thresholds ($\eta$)."
>
> **Empirical comparison with methods dropping non-tail elements:** We agree that comparing our approach with non-tail methods like (Finlayson et al., 2024) and Typical sampling [1] would be valuable. We are currently conducting these experiments.
>
> We have obtained results for Typical sampling.
>
> In GPT-2 Small:
>
> | method | perplexity | repetition | MAUVE |
> | ------- | ----------- | ----------- | --------- |
> | Typical | 10.785 | 0.019 | 0.888 |
> | Game | 23.590 | 0.002 | 0.926 |
> | Human | 21.559 | 0.002 | --- |
>
> In GPT-2 Medium:
>
> | method | perplexity | repetition | MAUVE |
> | ------- | ----------- | ----------- | --------- |
> | Typical | 9.018 | 0.0098 | 0.923 |
> | Game | 17.499 | 0.002 | 0.945 |
> | Human | 15.923 | 0.002 | --- |
>
> In GPT-2 Large:
>
> | method | perplexity | repetition | MAUVE |
> | ------- | ----------- | ----------- | --------- |
> | Typical | 6.590 | 0.012 | 0.924 |
> | Game | 15.458 | 0.001 | 0.947 |
> | Human | 13.755 | 0.002 | --- |
>
> In GPT-2 XL:
>
> | method | perplexity | repetition | MAUVE |
> | ------- | ----------- | ----------- | --------- |
> | Typical | 7.342 | 0.0108 | 0.932 |
> | Game | 11.333 | 0.003 | 0.958 |
> | Human | 12.319 | 0.002 | --- |
>
> We also experimented on larger models as suggested by other reviewers. In GPT-J-6B:
>
> | method | perplexity | repetition | MAUVE |
> | ------- | ----------- | ----------- | --------- |
> | Typical | 8.747 | 0.016 | 0.769 |
> | Game | 19.729 | 0.002 | 0.833 |
> | Human | 9.95 | 0.002 | --- |
>
> And we are evaluating Llama-2-7B models and will update you once ready.
>
> Our method has advantages over Typical sampling in open-ended tasks. Experiments on (Finlayson et al., 2024) are in progress. We will share the completed results later.
>
> **Interpreting the quantity $C_{I-1}$:** Let us clarify how $C_{I-1}=-\log W+\epsilon/W$ influences the growth of the support set {$1,...,I-1$}. An element $I$ is added to the support if its surprisal is less than $H(p_{1:I-1})+C_{I-1}$, where $W$ represents the total probability mass of $p_{1:I-1}$.
>
> While $C_{I-1}$ need not dominate the entropy term, its magnitude varies inversely with $W$. This creates a dynamic threshold: when $I$ is small and $W$ is low, the large value of $C_{I-1}$ encourages support expansion. As $I$ increases and $W$ grows, $C_{I-1}$ decreases to $\epsilon$, making the entropy $H(p_{1:I-1})$ more relevant in constraining further support growth.
>
> **Adaptation to other types of distances:** The choice of total variation distance is crucial for obtaining closed-form solutions. While our framework could theoretically extend to KL divergence or cross entropy, these metrics present significant analytical challenges.
>
> Under TV distance, despite the non-convexity of the $p$-minimization, the polytope structure of the feasible region enables characterization of the minimizer (We also discussed this in the previous part). In contrast, with KL divergence, the feasible region becomes
>
> {$p:p^\top\log p-p^\top\log\hat p\leq \epsilon$} $\cap$ {$p:p^\top1=1$}.
>
> This geometry is more complicated. As a proposal, it can be approached by introducing the transformation $z_i=\log p_i,z_i\leq0$, yielding:
>
> $$\min_q\max_{z\in Z}-q^\top z=\min_q h_Z(-q),$$
>
> where $h_Z$ is the support function [2] of the non-convex set
>
> $Z=${$z:\sum e^{z_i}(z_i-\log\hat p_i)\leq\epsilon,\sum e^{z_i}=1$}.
>
> While the $q$-optimization part remains convex (because $h_Z$ is always convex [2]), the $p$-optimization part is not straightforward because characterizing $h_Z(-q)$ is challenging due to the complicated geometry of $Z$. This problem itself may be an interesting future direction, and we are happy to add these discussions to our revision.
>
> ## Finally,
>
> Thank you again for your comprehensive and helpful review. Based on your feedback, we are now working on a revised version and will update you before the end of discussion period. During this time, please also let us know if there are any further questions.
>
> **References**
>
> [1] Clara Meister, Tiago Pimentel, Gian Wiher, and Ryan Cotterell. "Locally typical sampling." *Transactions of the Association for Computational Linguistics*, 11:102–121, 2023.
>
> [2] Stephen Boyd and Lieven Vandenberghe. *Convex Optimization*. Cambridge University Press, 2004.

---

> ### Author Response · Authors · 2024-11-26
>
> We would like to update you on the experiment results of BA sampling (Finlayson et al., 2024) in GPT-2 models, GPT-J-6B, and Llama-2-7B.
>
> In GPT-2 Small:
>
> | method | perplexity | repetition | MAUVE |
> | ------- | ----------- | ----------- | --------- |
> | BA | 29.610 | 0.002 | 0.917 |
> | Typical | 10.785 | 0.019 | 0.888 |
> | Game | 23.590 | 0.002 | 0.926 |
> | Human | 21.559 | 0.002 | --- |
>
> In GPT-2 Medium:
>
> | method | perplexity | repetition | MAUVE |
> | ------- | ----------- | ----------- | --------- |
> | BA | 24.119 | 0.0006 | 0.933 |
> | Typical | 9.018 | 0.0098 | 0.923 |
> | Game | 17.499 | 0.002 | 0.945 |
> | Human | 15.923 | 0.002 | --- |
>
> In GPT-2 Large:
>
> | method | perplexity | repetition | MAUVE |
> | ------- | ----------- | ----------- | --------- |
> | BA | 15.223 | 0.0008 | 0.954 |
> | Typical | 6.590 | 0.012 | 0.924 |
> | Game | 15.458 | 0.001 | 0.947 |
> | Human | 13.755 | 0.002 | --- |
>
> In GPT-2 XL:
>
> | method | perplexity | repetition | MAUVE |
> | ------- | ----------- | ----------- | --------- |
> | BA | 13.495 | 0.001 | 0.946 |
> | Typical | 7.342 | 0.0108 | 0.932 |
> | Game | 11.333 | 0.003 | 0.958 |
> | Human | 12.319 | 0.002 | --- |
>
> In GPT-J-6B:
>
> | method | perplexity | repetition | MAUVE |
> | ------- | ----------- | ----------- | --------- |
> | BA | 12.307 | 0.002 | 0.826 |
> | Typical | 8.747 | 0.016 | 0.769 |
> | Game | 19.729 | 0.002 | 0.833 |
> | Human | 9.95 | 0.002 | --- |
>
> In Llama-2-7B:
>
> | method | perplexity | repetition | MAUVE |
> | ------- | ----------- | ----------- | --------- |
> | BA | 8.375 | 0.002 | 0.890 |
> | Typical | 5.156 | 0.209 | 0.719 |
> | Game | 14.000 | 0.128 | 0.858 |
> | Human | 6.469 | 0.002 | --- |
>
> BA sampling has comparable performance with our method. As a strategy that drops non-tail elements, it scores higher in Llama-2-7B and GPT-2-Large, where our method is at the second place. Our method achieves a higher score in the other models. We also remark that BA involves matrix factorization and operates at a higher computation cost.
>
> We have included these results in our updated submission file. Thank you again for your review, and please let us know if there are further questions.

---

> > ### Comment · Reviewer_1fhp · 2024-11-26
> > **Rebuttal response**
> >
> > Thank you for the thorough response and integrating my comments!
> >
> > Following the updates, the paper seems more readable to me and my concerns were addressed.
> >
> > Since it is tricky to properly measure empirical quality of text (as a consequence of the limitations of mauve as a metric), the direct practical impact may be more limited than it appears to be and this is my main concern. However, as is, the paper does the most one could reasonably ask for to empirically validate the proposed method. Furthermore, I believe there is a lot of value in phrasing the problem this way and introducing this framework as far as future works are concerned. Hence, I decided to raise my score.

---

> ### Author Response · Authors · 2024-11-27
>
> We are very grateful for your positive evaluation and constructive comments on our submission.

---

### Official Review · Reviewer_Tan5 · 2024-11-07

**Soundness:** 2
**Presentation:** 2
**Contribution:** 2
**Rating:** 5
**Confidence:** 2

**Summary:**

This paper primarily focuses on text generation for language models.
Firstly, the author presents a novel minimax optimization problem for text generation tasks, where the min-player aims to minimize the worst-case negative cross-entropy.
Subsequently, it is demonstrated that the proposed minimax optimization problem can be interpreted as a likelihood maximization problem with implicit regularization.
Finally, the algorithm for solving the minimax optimization problem is introduced and its performance is empirically evaluated through experiments using GPT-2 models.

**Strengths:**

* The problem is appropriately motivated. Considering decoding strategies from a theoretical standpoint is essential for the research community.
* The algorithm proposed to solve the suggested minimax optimization problem appears to be easily implementable in practice.

**Weaknesses:**

I'm wondering to what extent this study provides new insights for existing decoding strategies that have achieved empirical success.
The author states "To resolve this dichotomy, this paper aims to propose a comprehensive theoretical framework of text generation" in the introduction section, meaning that one of the main goals of this paper is to theoretically explain the success of these existing strategies.
However, I'm not convinced that the proposed framework can replicate the widely recognized Nucleus sampling and Top-k sampling strategies simply by selecting an objective function $f$.
While it's demonstrated that temperature sampling can be recovered by appropriately choosing the function $f$, I'm interested in knowing if other strategies can also be replicated.

Moreover, since I'm not an expert in this research field, I'm not sure whether experiments using only GPT-2 models are sufficient for validating the proposed framework.
At the very least, it should be discussed whether the scale of the experiments conducted in this study is not insufficient compared to the current standards.

**Questions:**

My main concerns and questions are outlined in Weaknesses.
There is a typo in the introduction section:
* (line 55-56) "From a statistical perspective" is repeated.

---

> ### Author Response · Authors · 2024-11-20
>
> We thank the reviewer for their thoughtful feedback regarding the theoretical interpretability and experimental validation of our framework. We address each point in detail below.
>
> **Recovering Top-k or Nucleus sampling:**
>
> Both Top-k and Nucleus sampling follow a two-step process, first truncating the probability distribution, then rescaling the remaining probabilities. The key difference lies in how they determine their truncation thresholds. Similarly, other methods like $\eta$-sampling and Mirostat sampling follow this same truncation-rescaling pattern, each with their own threshold selection criteria.
>
> Rather than replicating **one single strategy** exactly, our primary contribution is understanding the fundamental design principles behind this **broad family of strategies** -- specifically, why do all of them follow a truncation-rescaling process?
>
> By setting $f(x)=\log x$ in our framework, we show that our approach naturally recovers a sampling method based on truncation and rescaling (Theorem 4.4 and Corollary 4.5). Although our method determines the truncation threshold differently from Top-k or Nucleus sampling, our framework provides key theoretical insights:
>
> 1. **Truncation** emerges as optimal because minimax robustness implicitly requires $\ell_{\infty}$ regularization.
> 2. **Uniform rescaling** of the remaining components is effective because it approximates the optimal solution in first order.
>
> Temperature rescaling, instead of uniform rescaling, is obtained when we change the objective $f$ from $\log$.
>
> **Experiments on GPT-2 vs larger models:** We agree that it is better to incorporate experiments on larger models. Following your suggestions, we have obtained results on GPT-J-6B model:
>
> | method | perplexity | repetition | MAUVE |
> | --- | --- | --- |--- |
> | Game | 19.729 | **0.002** | **0.833** |
> | Nucleus | 23.149 | **0.002** | 0.806 |
> | Greedy | 2.603 | 0.731 | 0.043 |
> | Pure | 27.327 | 0.001 | 0.798 |
> | Contrastive | 6.573 | 0.007 | 0.715 |
> | Typical | **8.747** | 0.016 | 0.769 |
> | Human | *9.95* | *0.002* | --- |
>
> Here, Contrastive and Typical are new methods suggested by other reviewers. Our method still maintains an edge in terms of having the highest MAUVE score.
>
> We are also evaluating Llama-2-7B model and will update you once it is ready.
>
> At the same time, we would like to clarify that recent works on decoding methods, such as [1], also only evaluated GPT-2 models. In the initial version of our paper, we aimed to follow the existing experiment settings.
>
> Thank you again for your comprehensive and helpful review. Based on your feedback, we are now working on a revised version and will update you before the end of discussion period. During this time, please also let us know if there are any further questions.
>
> **References**
>
> [1] Finlayson M, Hewitt J, Koller A, Swayamdipta S, Sabharwal A. "Closing the curious case of neural text degeneration." *The Twelfth International Conference on Learning Representations (ICLR)*, 2024.

---

> ### Author Response · Authors · 2024-11-26
>
> We would like to update you on the experiment results of Llama-2-7B models.
>
> | method | perplexity | repetition | MAUVE |
> | --- | --- | --- | --- |
> | Game | 14.000 | 0.128 | 0.858 |
> | Nucleus | 31.125 | 0.154 | 0.853 |
> | Contrastive | 5.375 | 0.236 | 0.702 |
> | Typical | 5.516 | 0.209 | 0.719 |
> | BA | 8.375 | 0.002 | 0.890 |
> | Greedy | 3.109 | 0.457 | 0.537 |
> | Pure | 44.500 | 0.016 | 0.813 |
> | Human | 6.469 | 0.002 | --- |
>
> Here, Contrastive [1], Typical [2], and BA [3] are decoding methods suggested by other reviewers. Our method scores the second highest in Llama-2-7B, next to BA, and maintains an edge over other methods like Nucleus sampling. We also remark that BA involves matrix factorization and operates at a higher computation cost than our method. The results show a consistent performance of our method in larger models such as GPT-J-6B (in our previous reply) and Llama-2-7B.
>
> We have included these results in our updated submission file. Thank you again for your review, and please let us know if there are any further questions.
>
> **References**
>
> [1] Yixuan Su, Tian Lan, Yan Wang, Dani Yogatama, Lingpeng Kong, and Nigel Collier. A contrastive framework for neural text generation. In Advances in Neural Information Processing Systems, 2022.
>
> [2] Clara Meister, Tiago Pimentel, Gian Wiher, and Ryan Cotterell. Locally typical sampling. Transactions of the Association for Computational Linguistics, 11:102–121, 2023.
>
> [3] Matthew Finlayson, John Hewitt, Alexander Koller, Swabha Swayamdipta, and Ashish Sabharwal. Closing the curious case of neural text degeneration. In The Twelfth International Conference on Learning Representations, 2024.

---

> > ### Comment · Reviewer_Tan5 · 2024-11-27
> >
> > Thank you for your response and for providing additional experimental results on larger models.
> >
> > I maintain the belief that it is crucial to discuss why the proposed framework can replicate some sampling methods while it fails to do so with others.
> > Specifically, Nucleus sampling exhibited superior performance in your experiments, which leads me to question what unique features nucleus sampling possesses that are unattainable by the proposed framework.
> > As a result, it would be beneficial to at least discuss the significant differences between the proposed and Nucleus sampling methods.

---

> ### Author Response · Authors · 2024-11-27
>
> Thank you for your reply.
>
> First, we would like to clarify that Nucleus sampling is not superior in our experiments (Table 1). Nucleus only performs better than our method in GPT-2 Large, and fails to do so in all the remaining models: Llama-2-7B, GPT-J-6B, GPT-2 XL, GPT-2 Medium, and GPT-2 Small.
>
> Yet we understand it is important to further compare our method with Nucleus sampling. The only difference lies in how each method decides the truncation threshold. In Nucleus sampling, the truncation threshold is $\max${$I:\sum_{i=1}^{I}p_i<P$}. In our method, it is $\max${$I:\sum_{i=1}^{I-1}p_i\log(p_i/p_I)<\epsilon$}. The former is based on probability while the latter has an interpretation from information theory, which we commented at around line 375.
>
> Due to the fact that $f(I)=\sum_{i=1}^{I-1}p_i\log(p_i/p_I)$ is increasing in $I$, we can still replicate Nucleus by setting an appropriate $\epsilon$, depending on $P$ (the hyperparameter of Nucleus) and the probability distribution $p_i$. Doing so, we will need an adaptive $\epsilon$ that changes at every decoding step. We have added these discussions to our paper at around line 383. To reiterate, our method comes from theoretical optimality results, and also performs consistently better than Nucleus in experiments.

---

### Author Response · Authors · 2024-11-21
**We have updated our submission**

Dear reviewers,

Thank you very much for your review. We have made an update to our pdf file submission based on your feedback to improve our presentation and include some new experiment results. The changes reflect our individual replies below.

As we are still working on extra experiments and polishing the presentation further, we anticipate to make one more update by the end of the discussion period. We hope that our current update can help the reviewers know about our progress and facilitate our discussion.

Please let us know if there are further questions and feedback.

Authors

---

### Author Response · Authors · 2024-11-26
**We have updated our submission**

Dear reviewers,

Thank you very much for your review. We have updated our submission to include new experiment results on Llama-2-7B and BA sampling method (Finlayson et al., 2024) and to continue polishing our presentation, as suggested by the reviews.

Please let us know if you have any other questions or feedback. We look forward to engaging in further discussions with you.

Authors

---

### Meta-Review · Area_Chair_B8Hi · 2024-12-21

**Metareview:**

The paper introduces a game-theoretic framework, coined "Decoding Game", which models text generation as a two-player zero-sum game. The framework is used to explain methods such as Top-k and Nucleus sampling as approximations to the optimal strategy in the decoding game, giving a common theoretical grounding to these approaches. Empirical evaluation on several language models, including GPT-2, GPT-J-6B, and Llama-2-7B, and showing competitive performance according to various metrics, complements the submission.

Some reviewers seemed initially skeptical regarding the potential for practical relevance of the results and certain details of the experimental setup. However, these concerns seem to have been largely resolved through author rebuttals and updates to the submission. The addition of experiments with larger models/baselines was particularly well-received. There appears to be positive consensus towards the value of the paper.

For that, an acceptance decision is recommended.

**Additional Comments On Reviewer Discussion:**

See above.

---

### Decision · Program_Chairs · 2025-01-22

Accept (Poster)